# Self-training Avoids Using Spurious Features Under Domain Shift

**Yining Chen**[*], **Colin Wei**[*], **Ananya Kumar, Tengyu Ma**
Department of Computer Science
Stanford University
{cynnjjs, colinwei, ananya1, tengyuma}@stanford.edu

## Abstract

In unsupervised domain adaptation, existing theory focuses on situations where the source and target domains are close. In practice, conditional entropy minimization and pseudo-labeling work even when the domain shifts are much larger than those analyzed by existing theory. We identify and analyze one particular setting where the domain shift can be large, but these algorithms provably work: certain spurious features correlate with the label in the source domain but are independent of the label in the target. Our analysis considers linear classification where the spurious features are Gaussian and the non-spurious features are a mixture of log-concave distributions. For this setting, we prove that entropy minimization on unlabeled target data will avoid using the spurious feature if initialized with a decently accurate source classifier, even though the objective is non-convex and contains multiple bad local minima using the spurious features. We verify our theory for spurious domain shift tasks on semi-synthetic Celeb-A and MNIST datasets. Our results suggest that practitioners collect and self-train on large, diverse datasets to reduce biases in classifiers even if labeling is impractical.

## 1   Introduction

Reliable machine learning systems need to generalize to test distributions that are different from the training distribution. However, the test performance of machine learning models often significantly degrades as the test domain drifts away from the training domain. Various approaches have been proposed to adapt the models to new domains [37, 10, 38] but theoretical understanding of these algorithms is limited. Prior theoretical works focus on settings where the target domain is sufficiently close to the source domain [5, 29, 17, 34]. To theoretically study realistic scenarios where domain shifts can be much larger, we need to leverage additional structure of the shifts.

Towards this goal, we propose to study a particular structured domain shift for which unsupervised domain adaptation is provably feasible: in the source domain, a subset of "spurious" features correlate with the label, whereas in the unlabeled target data, these features are independent of the label. In real-world training data, these spurious correlations can occur due to biased sampling or artifacts in crowd-sourcing [14]. For example, we may have a labeled dataset for recidivism prediction where race correlates with recurrence of crime due to sample selection bias, but this correlation does not hold on the population. Models which learn spurious correlations can generalize poorly on population data which does not have these biases [23]. In these settings, it could be impractical to acquire labels for an unbiased sub-sample of the population, but unlabeled data is often available.

We prove that in certain settings, perhaps surprisingly, self-training on *unlabeled* target data can avoid using these spurious features. Our theoretical results apply to two closely-related popular algorithms:

---

[*]Equal Contribution

self-training [20] and conditional entropy minimization [13]. In practice, self-training has achieved competitive or state-of-the-art results in unsupervised domain adaptation [22, 44, 30], but there are few theoretical analyses of self-training when there is domain shift.

Our theoretical setting and analysis are consistent with recent large-scale empirical results by Xie et al. [42], which suggest that self-training on a more diverse unlabeled dataset can improve the robustness of a model, potentially by avoiding using spurious correlations. These results and our theory help emphasize the value of a large and diverse unlabeled dataset, even if labeled data is scarce.

Formally, we assume that each input consists of two subsets of features, denoted by $x_1$ and $x_2$. $x_1$ is the "signal" feature that determines the label $y$ in the target distribution. $x_2$ is the spurious feature that correlates with the label $y$ in the source domain, but $x_2$ is independent of $(x_1, y)$ in the target domain. For a first-cut result, we consider binary classification and linear models on the features $(x_1, x_2)$, where the spurious feature $x_2$ is a multivariate Gaussian and $x_1$ is a mixture of log-concave distributions. We aim to show that, initialized with some classifier trained on the source data, self-training on the unlabeled target will remove usage of the spurious feature $x_2$.

A challenge in the analysis is that self-training on an unlabeled loss can possibly harm, rather than help, target accuracy by amplifying the mistakes of source classifier (see Section 3.1). The classical idea of co-training [8] deals with this by assuming the mistakes of the classifier are independent of $x$, reducing the problem to learning from noisy labels. However, in our setting the source classifier makes biased mistakes which depend on $x$, and self-training potentially reinforces these biases if there are no additional assumptions. For example, we require initialization with a decently accurate source classifier, and we empirically verify the necessity of this assumption in Section 5.

Our main contribution (Theorem 3.1) is to prove that self-training and conditional min entropy using finite unlabeled data converge to a solution that has 0 coefficients on the spurious feature $x_2$, assuming the following: 1. the signal $x_1$ is a mixture of well-separated log-concave distributions and 2. the initial source classifier is decently accurate on target data and avoids relying too heavily on the spurious feature. In a simpler setting where $x_1$ is a univariate Gaussian, we show that self-training using a decently accurate source classifier converges to the Bayes optimal solution (Theorem 3.2).

We run simulations on semi-synthetic colored MNIST [19] and celebA [21] datasets to verify the insights from our theory and show that they apply to multi-layer neural networks and datasets where the spurious features are not necessarily a subset of the input coordinates (Section 5).

## 1.1 Related Work

**Self-training** methods have achieved state-of-the-art results for semi-supervised learning [42, 32, 20], adversarial robustness [22, 44, 30], and unsupervised domain adaptation [22, 44, 30], but there is little understanding of when and why these methods work under domain shifts. Two popular forms of self-training are pseudolabeling [20] and conditional entropy minimization [13], which have been observed to be closely connected [2, 20, 30, 7]. We show that our analysis applies to both entropy minimization and a version of pseudo-labeling where we initialize the student model with the teacher model and re-label after each gradient step (Proposition D.1).

Kumar et al. [18] examine self-training for domain adaptation, but under strong assumptions: that $P(X|Y)$ is an isotropic Gaussian, that entropy minimization converges to the nearest local minima, and infinite unlabeled data. They use a symmetry argument that requires all these assumptions. In our setting, the signal $x_1$ can be a mixture of many log-concave, log-smooth distributions, and we show that self-training does in fact converge with only finite unlabeled data, even though the loss landscape is non-convex. These require new, more general, proof techniques.

**Domain adaptation and semi-supervised learning theory**: Importance weighting [29, 17, 34] is a popular way to deal with *covariate shift* but these methods assume that $P(Y \mid X)$ is the same for the source and target, which may not hold when there are spurious correlations in the source but not target. Additionally, sample complexity bounds for importance weighting depend on the expected density ratios between the source and target, which can often scale exponentially in the dimension. Our finite sample guarantees only depend on properties of the target distribution (assuming a decently accurate source classifier) and are agnostic to this density ratio. The theory of $H\Delta H$-divergence lower bounds target accuracy of a classifier in terms of source accuracy if some distance between the domains is small [6]; Zhang et al. [43] extend this distance measure to multiclass classification. In

contrast, we show self-training can improve accuracy under our structured domain shift, even when the shift is potentially large. Other theoretical papers on semi-supervised learning focus on analyzing when unlabeled data can help, but do not analyze domain shift [27, 31, 5, 4].

Co-training [8] is an algorithm that can leverage unlabeled data when the input features can be split into $(x_1, x_2)$ that are conditionally independent given the label. Co-training assumes this grouping is known a-priori, and that either group can be used to predict the label accurately. In our setting, the spurious feature cannot be used to predict the label accurately in the target domain, and the algorithm does not have access to the grouping between spurious and signal features.

**Spurious and non-robust features.** Many works seek to identify causal features invariant across domains [3, 26, 15]. Spurious features are also related to adversarial examples, which can possibly be attributed to non-robust features that can predict the label but are brittle under domain shift [16].

A number of papers theoretically analyze the connection between adversarial robustness and accuracy or generalization for linear classifiers in simple Gaussian settings [36, 28, 39]. Carmon et al. [9] show that self-training on unlabeled data can improve adversarially robust generalization for linear models in a Gaussian setting. Though these research questions are orthogonal to ours, one technical contribution of our work is that our analysis extends to more general distributions than Gaussians.

**Fairness.** Spurious correlations in datasets can lead to unfair predictions when protected attributes are involved. Our work shows that self-training can potentially employ unlabeled population samples to overcome bias in labeled data [12, 35].

## 2 Setup

**Model.** We consider a linear model $\widehat{y} = w^\top x$ where $w = (w_1, w_2)$ and $x = (x_1, x_2)$ with $w_1, x_1 \in \mathbb{R}^{d_1}$ and $w_2, x_2 \in \mathbb{R}^{d_2}$. We assume that the spurious features $x_2$ have Gaussian distribution with covariance $\Sigma_2 \succ 0$, so the target data $(x, y) \sim \mathcal{D}_{\text{tg}}$ is generated by

$$y \overset{\text{unif}}{\sim} \{\pm 1\}, \text{and } x_1 \sim \mathcal{D}_{\text{tg},1}(\cdot|y)$$
$$x_2 \sim \mathcal{N}(\vec{0}, \Sigma_2), \Sigma_2 \in \mathbb{R}^{d_2 \times d_2} \tag{2.1}$$

for some distribution $\mathcal{D}_{\text{tg},1}$ over $\mathbb{R}^{d_1}$. Note that $x_2$ is a spurious feature because it is independent of the label $y$. Our results and analysis also transfer to a "scrambled setup" [3] where we observe $z = \mathcal{S}x$ for some rotation matrix $\mathcal{S} \in \mathbb{R}^{(d_1+d_2) \times (d_1+d_2)}$. This follows as a direct consequence of the rotational invariance of the algorithm (2.3) and our assumptions.

**Min-entropy objective.** The min-entropy objective on a target unlabeled example is defined as $\ell_{ent}(w^\top x)$ where $\ell_{ent}(t) = H((1 + \exp(-t))^{-1})$ and $H$ is the binary entropy function.

For mathematical convenience, we consider an approximation $\ell_{exp}(t) = \exp(-|t|)$, which is commonly used in the literature for studying the logistic loss [33]. $\ell_{exp}$ approximates $\ell_{ent}$ up to a constant factor and exhibits the same tail behavior (Figure 10). We experimentally validate in Section E.5 that training using $\ell_{exp}$ achieves the same effect for the algorithms we analyze. The population unlabeled objective on the target distribution that we consider is

$$L(w) \triangleq \mathop{\mathbb{E}}_{x \sim \mathcal{D}_{\text{tg}}} \ell_{exp}(w^\top x) \tag{2.2}$$

where $\mathcal{D}_{\text{tg}}$ denotes the distribution in the target domain. We mainly focus on analyzing the population loss for simplicity, but in our main results (Theorems 3.1 and 3.2) we also give finite-sample guarantees. We analyze the following equivalent algorithms for self-training.

**Entropy minimization.** We initialize $w$ from a source classifier $w^{\text{S}}$ and run projected gradient descent on the entropy objective:[2]

$$w^0 = w^{\text{S}} \text{ and } w^{t+1} = \frac{w^t - \eta \nabla L(w^t)}{\|w^t - \eta \nabla L(w^t)\|_2} \tag{2.3}$$

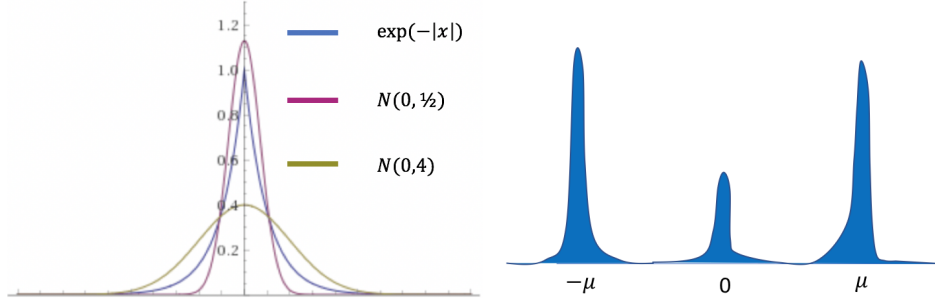

Figure 1: Cases where entropy minimization fails to remove $w_2$. **Left (Example 3.1):** When $w_1 = 0$, $w^\top x$ is distributed as a Gaussian. Increasing $w_2$, which increases this variance (e.g, going from the purple to green curve) decreases $L(w)$ by forcing every prediction further from 0. This means that entropy minimization causes reliance on the spurious feature to *increase*. **Right (Example 3.2):** Distribution of $w_1^\top x_1$ in a hard case for general distributions. If there is a lot of mass of $w_1^\top x_1$ concentrated near the boundaries (i.e, $\pm\mu$ for some large $\mu \to \infty$) and a small amount of mass near 0, the loss could be small but the classifier will not want to shrink $\|w_2\|_2$.

**Pseudo-labeling.** We consider a variant of pseudo-labeling where we label the target data using the classifier from the previous iteration and run projected gradient descent on the supervised loss

$$L_{pseudo}^{t+1}(w) \triangleq \underset{x \sim \mathcal{D}_{\text{tg}}}{\mathbb{E}} \ell_{exp}(w^\top x, y^t) \tag{2.4}$$

where $y^t = \text{sgn}\left(w^{t^\top} x\right)$ and $\ell_{exp}(t, y) = \exp\left(-ty\right)$. The algorithm is the same as 2.3 with $L(w)$ replaced by $L_{pseudo}^{t+1}(w)$. Note that this is different from some versions of pseudo-labeling, which train for many rounds of gradient descent before re-labeling. We observe that the two algorithms above are equivalent because the iterates are the same (see Section D.3 for the formal proof).

## 3 Overview of Main Results

We would like to show that entropy minimization (2.3) drives the spurious feature $w_2$ to 0. However, this is somewhat surprising and challenging to prove because nothing in the loss or algorithm explicitly enforces a decrease in $\|w_2\|_2$. Indeed, without additional assumptions on the target distribution $\mathcal{D}_{\text{tg}}$ and the initial source classifier $w^{\text{S}}$, we show that entropy minimization can actually cause $\|w_2\|_2$ to increase because self-training can reinforce existing biases in the source classifier.

Examples 3.1 and 3.2 highlight cases where entropy minimization can fail, which motivates our assumptions of separation (Assumption 3.1) and that $x_1$ is a mixture of *sliced log concave* distributions. Under these assumptions, our main Theorem 3.1 shows that entropy minimization (2.3) initialized with a decently accurate source classifier drives the coefficient of the spurious feature, $w_2$, to 0. For a simpler Gaussian setting, Theorem 3.2 shows that entropy minimization with a sufficiently accurate source classifier converges to the Bayes optimal classifier.

### 3.1 Failure cases of self-training

We highlight cases where self-training increases reliance on the spurious features, justifying our assumptions in Section 3.2.

**Example 3.1** (No contribution from signal, i.e. $w_1^\top x_1 = 0$.)**.** *See Figure 1 (Left). For simplicity, suppose that $d_1 = d_2 = 1$, and suppose that $w_1 = 0$, so the signal feature is not used. In this case, increasing $|w_2|$ drives every prediction further from 0, decreasing the expected loss $L(w)$. Thus, in this example the min-entropy loss actually encourages the weight on the spurious feature, $|w_2|$, to increase. Note that this is not trivially true when $w_1$ is nonzero.*

In a realistic scenario, it's unlikely that $w_1^\top x_1 = 0$ for all examples because then the source accuracy on the target domain is very poor. So a priori, if we assume the source accuracy is decent (which implies $L(w^{\text{S}})$ is small), we may avoid the pathological case above. However, this is not sufficient.

**Example 3.2** (Initial $L(w^{\text{S}})$ is small, but self-training still increases $\|w_2\|_2$.)*. See Figure 1 (Right). Suppose that restricting to the signal feature, we have a mixture of perfectly and extremely confidently predicted examples, and a small amount of unconfident examples as in Example 3.1. The majority group of confident examples is already perfectly predicted with no incentive to remove $w_2$ (because the loss gradient is near 0), and the minority group encourages $\|w_2\|_2$ to increase as in Example 3.1, so the overall effect is for $\|w_2\|_2$ to increase though $L(w)$ is small.*

For self-training to succeed, the correctly and confidently predicted examples must help remove the spurious features. As demonstrated above, this requires some continuum between confidently and unconfidently predicted examples. This motivates the log-concavity and smoothness assumptions, which guarantees that the sample distribution is not supported on too many extremely isolated clusters.

## 3.2 Mixtures of log-concave and log-smooth distributions

To avoid the failure cases above, we make realistic assumptions which are plausible in real-world data distributions. We start by defining a variant of log-concave and log-smooth distributions.

**Definition 3.1** (sliced log-concavity, log-smoothness)*. A distribution over $\mathbb{R}^d$ with density $p$ is $\alpha$-log-concave for $\alpha > 0$ if $\nabla^2 \log p(t) \preceq -\alpha \cdot I_{d \times d}$, and is $\beta$-log-smooth if $\|\nabla^2 \log p(t)\|_{\text{op}} \leq \beta$. A distribution $p$ over $\mathbb{R}^d$ is sliced $\alpha$-log-concave or sliced $\beta$-log-smooth if for any unit vector $v$, the random variable $v^\top x$ with $x \sim p$ is $\alpha$-log-concave or $\beta$-log-smooth, respectively.*

A 1 dimensional density that is not Gaussian which satisfies these assumptions is $p(x) \propto \exp(-x^2 + \cos x)$. This density is 1-log concave and 3-log smooth. Now we state our main assumption that $x_1$ consists of a mixture of sliced-log-concave and smooth distributions with sufficient separation.

**Assumption 3.1** (Separation assumption on the data)*. We assume that the distribution of $x_1$ in the target domain, denoted by $\mathcal{D}_{\text{tg},1}$, is a mixture of $K$ sliced $\alpha$-log-concave and $\beta$-log-smooth distributions. (The reader can think of $\alpha$, $\beta$ and $K$ as absolute constants for simplicity.) Let $\tau_1, \ldots, \tau_K$ denote the probability of each mixture and $\tau = \min_i \tau_K$. We assume that these mixtures are sufficiently separated in the sense that for scalar $\kappa$ (that depends on $\alpha$ and $\beta$), there exists $(w_1, 0) \in \mathbb{R}^{d_1 + d_2}$ such that $L((w_1, 0)) \leq \tau \kappa$.*

We formally define $\kappa$ in Section B.1. When $\alpha$ and $\beta$ are of constant scale, $\kappa$ is also a constant. Assumption 3.1 is always satisfied if the means of each mixture distribution in $\mathcal{D}_{\text{tg},1}$ are sufficiently bounded from 0. We can see why Assumption 3.1 is a separation condition by considering the case when there exists $(w_1, 0)$ with good classification accuracy on $x_1$. Obtaining good classification accuracy is only possible if the means of different classes are sufficiently far from 0 and also on opposite sides of 0, resulting in separation between the two classes.

The sliced log-concavity ensures that each mixture component of $w_1^\top x_1$ is uni-modal, with upper bound $\alpha$ on its "width". Likewise, the sliced log-smoothness condition ensures that each component is not too narrow. These conditions rule out the hard distribution in Figure 1 (right), as each of the three components change quite sharply, violating log-smoothness. Next, we assume the source classifier is decently accurate and has bounded usage of the spurious feature.

**Assumption 3.2** (Source classifier is decently accurate, doesn't rely too much on spurious features.)*. We assume that the source classifier has a significant mass in the space of the signal $x_1$ in the sense that $\|w^{\text{S}}_1\|_2 \geq 1/2$. We further assume that either $\Sigma_2$ is sufficiently small or $w^{\text{S}}_2$, the initialization in the spurious feature space, is sufficiently small, in the sense that*

$$\sigma^2 = w^{\text{S}}_2{}^\top \Sigma_2 w^{\text{S}}_2 \leq c \cdot \min\{1, \alpha/\beta^2, \beta^{-1}, (\beta/|\log \beta|)^{-1}\}$$

*for some sufficiently small universal constant $c$ (e.g., $c = 0.03$ can work.) Furthermore, we assume the source classifier $w^{\text{S}}$ has small entropy bound $L(w^{\text{S}}) \leq \tau \kappa$, where $\kappa$ is the constant in Assumption 3.1.*

The conditions on $\|w^{\text{S}}_1\|_2$ and $\sigma$ can be satisfied if $w^{\text{S}}_2$ is not too large. The following theorem shows that under our assumptions, entropy minimization succeeds in removing the spurious $w_2$.

**Theorem 3.1** (Main result)*. In the setting above, suppose Assumptions 3.1 and 3.2 hold and $L$ is smooth.[3] If we run Algorithm (2.3) initialized with $w^{\text{S}}$ with sufficiently small step size $\eta$, after*

$O(\log \frac{1}{\epsilon})$ *iterations, we will obtain* $\widehat{w}$ *with very small usage of spurious features, i.e.* $\|\widehat{w}_2\|_2 \le \epsilon$. *The same conclusion holds with probability* $1 - \delta$ *in the finite sample setting with* $O(\frac{1}{\epsilon^4} \log \frac{1}{\delta})$ *unlabeled samples from* $\mathcal{D}_{\text{tg}}$.

Above, the notation $O(\cdot)$ hides dependencies on $\alpha, \beta, \Sigma_2$, and other distribution-dependent parameters. The interpretation of Theorem 3.1 is that self-training can take a source classifier that has decent accuracy on the target and de-noise it completely, improving target accuracy by removing spurious extrapolations. The main proof of Theorem 3.1 is given in Section B. We provide proof intuitions in Section 4. In Section B.4, we show we can ensure convergence to an approximate local minimum of the objective $\min_{\|w_1\|_2 \le 1} L((w_1, 0))$ by adding Gaussian noise to the gradient updates.

**Special case: mixtures of Gaussians.** We provide a slightly stronger analysis of Algorithm 2.3 when the signal $x_1$ is a one-dimensional Gaussian mixture, i.e. we set $\mathcal{D}_{\text{tg},1}(\cdot|y) = \mathcal{N}(y\gamma, \sigma_1^2)$. Let $\tilde{\sigma}_{min}, \tilde{\sigma}_{max}$ denote the minimum and maximum eigenvalues of $\tilde{\Sigma} \triangleq \begin{pmatrix} \sigma_1^2 & 0 \\ 0 & \Sigma_2 \end{pmatrix}$.

We analyze a slightly more general variant of Algorithm 2.3 which projects to the $R$-norm ball rather than unit ball. We now show that starting from an initial source classifier with sufficiently high accuracy on the target domain, self-training will avoid using the spurious feature and converge to the Bayes optimal classifier. Note that this is a stronger statement than Theorem 3.1, which does not bound the final target accuracy of the classifier.

**Theorem 3.2.** *In the setting above, suppose we are given a classifier (trained on a source distribution)* $w^{\text{S}}$ *with* $\|w^{\text{S}}\| \le R$ *and 0-1 error on the target domain at most* $\rho = \frac{1}{2} \operatorname{erfc} \left( \frac{r(R\tilde{\sigma}_{max})}{\sqrt{2}R\tilde{\sigma}_{min}} \right)$.[4] *($r$ is a function as defined in Section A). Then Algorithm 2.3 converges to* $w^K$ *satisfying* $w_1^K \ge \sqrt{R^2 - \epsilon^2}$ *and* $\|w_2^K\|_2 \le \epsilon$ *within* $K = O(\log \frac{1}{\epsilon})$ *iterations. For the finite sample setting, the same conclusion holds with probability* $1 - \delta$ *using* $O(\frac{1}{\epsilon^4} \log \frac{1}{\delta})$ *samples.*

As above, $O(\cdot)$ hides dependencies in $R, \tilde{\sigma}_{min}, \tilde{\sigma}_{max}, \rho, d_2$. In particular, $w$ converges to $(R, 0)$, the classifier in $\{w : \|w\|_2 \le R\}$ with the best possible accuracy. The full proof is in Section A.

# 4   Overview of Analysis

We will summarize the key intuitions for proving Theorems 3.1 and Theorem 3.2. The main ingredient is to show that the min entropy objective encourages a decrease in $\|w_2\|_2$, as stated below:

**Lemma 4.1.** *In the setting of Theorem 3.2, suppose that the classifier* $w$ *has at most* $\rho$ *error on the target. Then* $\langle \nabla_{w_2} L(w), w_2 \rangle \ge 0$. *This same conclusion holds in the setting of Theorem 3.1 for any* $w$ *satisfying the conditions in Assumption 3.2.*

The consequence of Lemma 4.1 is that one step of gradient descent on the loss function $L(w)$ shrinks the norm of $w_2$. This leads to the conclusion of Theorems 3.1 and 3.2, modulo a few other nuances such as showing that the conditions of Lemma 4.1 hold for all the iterates, which is done inductively. We also show that $\|w_1\|_2$ increases after one gradient step (Lemma A.2), so the norm of $w_2$ still decreases after re-normalization. To prove Lemma 4.1, we first express the objective as follows:

$$L(w) = \mathbb{E}_{x_1} \left[ \mathbb{E}_{x_2} \ell_{exp}(w_1^\top x_1 + w_2^\top x_2) \right] \tag{4.1}$$

Note that $w_2^\top x_2$ has Gaussian distribution with mean zero and variance $\sigma^2 \triangleq w_2^\top \Sigma_2 w_2$. Let $g_\sigma(t) = \mathbb{E}_{z \sim \mathcal{N}(0,\sigma^2)}[\ell(t+z)]$ denote the convolution of $\ell_{exp}$ with $\mathcal{N}(0, \sigma^2)$. Then we can rewrite the loss as $L(w) = \mathbb{E}_{w_1^\top x_1}\left[g_\sigma(w_1^\top x_1)\right]$, so $\nabla_{w_2} L(w) = \frac{\partial L(w)}{\partial \sigma} \cdot \frac{\partial \sigma}{\partial w_2} = \frac{\partial L(w)}{\partial \sigma} \cdot 2\Sigma_2 w_2$, which implies $\langle \nabla_{w_2} L(w), w_2 \rangle = 2w_2^\top \Sigma_2 w_2 \frac{\partial}{\partial \sigma} L(w)$. As $\Sigma_2 \succ 0$, proving Lemma 4.1 is equivalent to proving $\frac{\partial}{\partial \sigma} L(w) \ge 0$. Letting $\mu \triangleq w_1^\top x_1$, we have $\frac{\partial}{\partial \sigma} L(w) = \mathbb{E}_\mu[q_\sigma(\mu)]$ where $q_\sigma(\mu) = \frac{\partial}{\partial \sigma} g(\mu)$. We now investigate when $q_\sigma(\mu) \ge 0$. As visualized in Figure 2, $q_\sigma(\mu) < 0$ for $\mu$ near 0.

**Case when** $\mu \gg \sigma$**:** Recall that $g_\sigma(\mu) = \mathbb{E}_{z \sim \mathcal{N}(\mu, \sigma^2)}[\ell_{exp}(z)]$ is the average of the entropy function over a Gaussian distribution. When $\mu$ is sufficiently large, most of the mass of the Gaussian

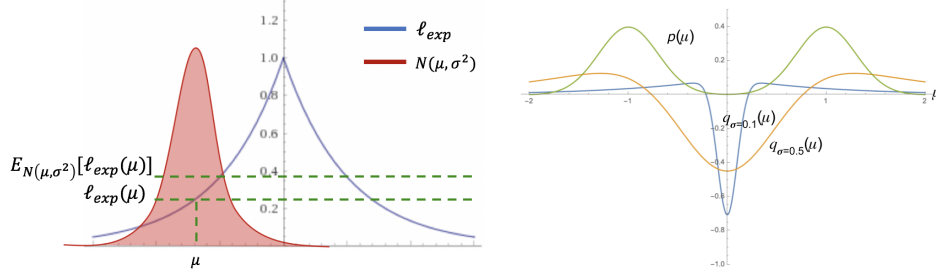

Figure 2: Analyzing dependence of $L$ on $\sigma$. **Left: $\mu \gg \sigma$.** A visual depiction of why $q_\sigma(\mu) > 0$ when $\mu \gg \sigma$. Conditioned on $\mu = w_1^\top x_1$, $w^\top x$ is Gaussian with mean $\mu$. As $\mu \gg \sigma$, most of its mass is in the region where $\ell_{exp}$ is convex. By Jensen's inequality, driving $\sigma$ to 0 decreases the loss in this region. **Right: plot of $q_\sigma(\mu)$.** The function $q_\sigma(\mu)$ will be convolved with $p$, the distribution over $\mu$. To guarantee $\mathbb{E}[q_\sigma(\mu)] \geq 0$, we would like to $\mu$ has as large amount of mass right to the positive root and left to the negative root of $q_\sigma(\cdot)$ as possible.

distribution $\mathcal{N}(\mu, \sigma^2)$ is on the positive side of the real line, where the function $\ell_{exp}$ is convex. For convex functions $f$, Jensen's inequality tells us $\mathbb{E}_{z \sim \mathcal{N}(\mu,\sigma^2)}[f(\mu)] > f(\mu)$ if $\sigma > 0$. As decreasing $\sigma$ decreases the expected loss, we can see that $q_\sigma(\mu) > 0$. This is visualized in Figure 2 (left).

**General case:** We plot $q_\sigma(\mu)$ as a function of $\mu$ for various choices of $\sigma$ in Figure 2 (right). We can see from the figure that for any $\sigma > 0$, there is a threshold $r(\sigma)$ (defined in (A.1)) such that for any $|\mu| > r(\sigma)$, $q_\sigma(\mu) > 0$. In Lemma A.4, we bound this value $r(\sigma)$ in terms of $\sigma$.

For the Gaussian setting, we can compute $\mathbb{E}_\mu[q_\sigma(\mu)]$ exactly and show it is positive for sufficiently accurate classifiers. For the general case (Theorem 3.1), it is difficult to bound this expectation because the expression for $q_\sigma(\mu)$ is complicated. Intuitively, our argument for why $\mathbb{E}_\mu[q_\sigma(\mu)] > 0$ is as follows: log-concavity and smoothness of each mixture component in $\mu$ ensures that the densities are uni-modal and do not change too fast. Thus, when $L(w)$ is sufficiently small, the mass of each component is spread over the real line, with most of the mass in the middle where $q_\sigma(\mu)$ is significantly positive, guaranteeing $\mathbb{E}_\mu[q_\sigma(\mu)] > 0$. To formalize this, we use a second order Taylor expansion of the log density of $\mu$ and bound the error incurred by the expansion using smoothness and concavity. This analysis is presented in Section B.

In Section C, we prove finite sample guarantees by showing that the gradient updates on the population and sample loss are similar ($\nabla \widehat{L}(w) \approx \nabla L(w)$ for all $w$, where $\widehat{L}(w)$ is the empirical loss).

# 5 Experiments

We validate our theory in a variety of empirical settings. We study a more general setting with nonlinear models where the signal $x_1$ and spurious feature $x_2$ are not distinct dimensions of the data. Using a semi-synthetic colored MNIST dataset, we verify that 1. self-training avoids using spurious features in a manner consistent with our theory and 2. as our theory predicts, self-training can harm performance when the source classifier is not sufficiently accurate. We also confirm our theoretical conclusions on a celebA dataset modified to have spurious correlations in training data but not in test.

Next, we investigate the connection between entropy minimization (2.3) and a variant of pseudo-labeling (2.4). We demonstrate that entropy minimization can converge to better target accuracy within a fixed wall clock time-budget, suggesting that practitioners may benefit from pseudo-labeling with more rounds and fewer epochs per round (Section E.3). In Section E.4 we verify that our conclusions hold for more common variants of pseudo-labeling in a toy Gaussian setting.

**Colored MNIST.** We create colored variants of the MNIST dataset [19] where the shape of the digit is the signal feature and the color is the spurious feature. In the first variant, denoted CMNIST10, there are 10 classes. Color correlates with the label in the source with probability $p = 0.95$, but is uncorrelated with the label in the target. In the second variant, denoted CMNIST2, we group digits into two classes: 0-4 and 5-9, which allows detailed investigation of our theory. Color correlates with the class label in the source but not in the target using a construction described in detail in Section E.1. We train 3-layer feed-forward network on the source, and use this to initialize entropy minimization (Algorithm 2.3) on unlabeled target data. Evaluation is performed on held-out target samples.

Table 1: Accuracy of models on the target before/after self-training, demonstrating that self-training can boost target accuracy under our structured domain shift. The exception is CMNIST10 with 0.97 probability of correlation between color and class. Here self-training decreases accuracy because initial accuracy is poor (only 72%), justifying our assumption of a decently accurate source classifier.

|  | CELEBA | CMNIST10 (P = 0.95) | CMNIST2 | CMNIST10 (P = 0.97) |
|---|---|---|---|---|
| TRAINED ON SOURCE | 81% | 82% | 94% | 72% |
| AFTER SELF-TRAINING | 88% | 91% | 96% | 67% |

Table 2: Number of test examples explainable by our theory. See text for definitions and interpretation.

|  | -/+ | +/- | +/+ | -/- | TOTAL |
|---|---|---|---|---|---|
| EXPLAINABLE | 271 | 45 | 8785 | 150 | 9251 |
| TOTAL | 349 | 86 | 9286 | 279 | 10000 |

**CelebA dataset.** Inspired by [15], we partition the celebA dataset [21] so that gender correlates perfectly with hair color in source data (Figure 7a) but not in the target (Figure 7b). We train a neural net to predict gender by first training on source data alone and then performing self-training with unlabeled target data. During self-training, we add the labeled source loss to the min-entropy loss on target data. (Section E.2 has more details.)

**Self-training improves target accuracy.** Table 1 shows that with a decently accurate source classifier, self-training on unlabeled target data leads to substantial improvements in the target domain. For example, on celebA the classifier achieves 81% accuracy before self-training and 88% after. This suggests that practitioners can potentially avoid overfitting to spurious correlations by self-training on large unlabeled datasets in the target domain.

**Self-training requires decent source classifier accuracy to succeed.** We test whether self-training is effective when the source classifier is bad by increasing the correlation between label and spurious color feature from 0.95 to 0.97 for CMNIST10. The resulting source classifier only obtains 72% initial accuracy on target data, which *drops* to 67% after self-training (see Table 1, last column, and plots in Section E.1). This shows that our assumption that the source classifier has to obtain non-trivial target accuracy (with bounded usage of the spurious feature) is also necessary in practice.

**Self-training reduces reliance on the spurious features.** In the CelebA experiment, test predictions corrected by self-training were mostly mistaken due to the spurious correlation. Figure 7c, a random sample of the corrected examples, consists of mostly blond females, non-blond males, and subjects with hats or irregular hairstyles.

For 2-class colored MNIST, let $\mu_S(x_1)$, $\sigma_S(x_1)$ denote the mean and standard deviation of the source classifier conditioned on grayscale image $x_1$, with color distributed independently of $x_1$. Define $\mu_T(x_1)$, $\sigma_T(x_1)$ similarly for the classifier after self-training. Our theory suggests that $\text{sgn}\, \mu_S(x_1) = \text{sgn}\, \mu_T(x_1)$, $|\mu_S(x_1)| < |\mu_T(x_1)|$, and $\sigma_S(x_1) > \sigma_T(x_1)$, and we say a test example is *explainable* by our theory if this holds. We divide the test examples into four categories: "-/+", "+/-", "+/+", "-/-", where, for example, "-/+" indicates source classifier was wrong but corrected by self-training. Table 2 summarizes the number of explainable examples in each category, showing that for the majority ($> 90\%$) of examples, entropy minimization works due to the reason we hypothesized. In Section E.1, we provide additional detailed analyses of the influence of the spurious color feature on the prediction before and after self-training.

# 6  Conclusion

We study the impact of self-training under domain shift. We show that when there are spurious correlations in the source domain which are not present in the target, self-training leverages the unlabeled target data to avoid relying on these spurious correlations. Our analysis highlights several conditions for self-training to work in theory, such as good separation between classes and a decently accurate source classifier. Our experiments support that 1) these theoretical requirements can capture the initial conditions needed for self-training to work and 2) under these initial conditions, self-training can indeed prevent the model from using spurious features in ways predicted by our theory. It is an interesting question for future work to explore other settings we can analyze with our framework.

## Broader Impact

Our work promotes robustness and fairness in machine learning. First, we study algorithms that make machine learning models robust when deployed in the real world. Second, our work addresses the scenario where the target domain is under-resourced and hence collecting labels is difficult. Third, our theoretical work guides efforts to mitigate dataset bias. We demonstrate that curating a diverse pool of unlabeled data from the true population can help combating existing bias in labeled datasets. We give conditions for when bias will be mitigated and when it will be reinforced or amplified by popular algorithms used in practice. We take a first step towards understanding and preventing the adverse effects of self-training.

## Acknowledgments and Disclosure of Funding

We are grateful to Rui Shu, Shiori Sagawa, and Pang Wei Koh for insightful discussions. The authors would like to thank the Stanford Graduate Fellowship program for funding. CW acknowledges support from a NSF Graduate Research Fellowship. TM is also partially supported by the Google Faculty Award, Stanford Data Science Initiative, and the Stanford Artificial Intelligence Laboratory.

## Footnotes

[2]We project to the unit ball for simplicity, as the loss $\ell_{exp}$ is not scale-invariant.

[3]As there is a discontinuity in $\frac{d}{dt} \ell_{exp}(t)$ at $t = 0$, we need to assume smoothness. This regularity condition is easy to satisfy; for example, it holds if $\mathcal{D}_{\text{tg},1}$ is a mixture of Gaussians.

[4] $\operatorname{erfc}(t) = \frac{2}{\pi} \int_t^\infty \exp(-x^2)dx$.

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
