[Supplementary Material]

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

# A  Warmup: Proofs for Gaussian Setting (Theorem 3.2)

We will define the function $r$ in Theorem 3.2 as follows:

$$r(\sigma) = \begin{cases} \sigma^2 + \sigma\sqrt{2\log\frac{4\sqrt{2}}{\sqrt{\pi}\sigma}}, & \text{if } 0 < \sigma \leq \frac{4\sqrt{2}}{\sqrt{\pi}}. \\ 2\sigma^2, & \text{if } \sigma > \frac{4\sqrt{2}}{\sqrt{\pi}}. \end{cases} \tag{A.1}$$

The algorithm that we consider is a variant of Algorithm 2.3 which more generally projects to the $R$-norm ball rather than unit ball: $w^{t+1} = R\frac{w^t - \eta\nabla L(w^t)}{\|w^t - \eta\nabla L(w^t)\|}$. Now define

$$a = \sqrt{2}R\tilde{\sigma}_{min}\,\mathrm{erfc}^{-1}(2\rho) = r(R\tilde{\sigma}_{max})$$
$$S = \{w : w_1\gamma \geq a, \|w\|_2 \leq R\}. \tag{A.2}$$

where the function $r$ is defined in Lemma A.4. We first observe that the condition that classifier $w$ has at most $1 - \rho$ error corresponds directly to $w \in \S$.

**Lemma A.1.** *In the setting of Theorem 3.2, suppose some classifier $w$ has at least $1 - \rho$ accuracy in the sense that $\Pr_{x,y}\left[\mathrm{sgn}(w^\top x) = y\right] \geq 1 - \rho$ and $\|w\|_2 \leq R$. Then $w_1\gamma \geq a$.*

*Proof of Lemma A.1.* Let $\tilde{\sigma}^2 = w^\top\tilde{\Sigma}w$. We have $R\tilde{\sigma}_{min} \leq \tilde{\sigma} \leq R\tilde{\sigma}_{max}$.

$$\Pr_{y\sim\{\pm1\},z\sim\mathcal{N}(\vec{0},\tilde{\sigma}^2)}\left[\mathrm{sgn}(yw_1\gamma + z) = y\right] \geq 1 - \rho$$
$$\iff \Pr_{z\sim\mathcal{N}(\vec{0},\tilde{\sigma}^2)}\left[w_1\gamma + z \geq 0\right] \geq 1 - \rho$$
$$\iff \mathrm{erfc}\left(\frac{w_1\gamma}{\sqrt{2}\tilde{\sigma}}\right) \leq 2\rho$$
$$\implies w_1\gamma \geq a$$

$\square$

Next, our proof of Theorem 3.2 will be based on the following two lemmas. The first lemma shows that $w_1$ is increasing, and the second shows that $\|w_2\|_2$ is decreasing at a fast enough rate.

**Lemma A.2.** *In the setting of Theorems 3.1 and 3.2, for any $w \in S$,*

$$\langle\nabla_{w_1}L(w), w_1\rangle < 0.$$

*Proof.* Recall the definition

$$g_\sigma(\mu) = \mathbb{E}_{z\sim\mathcal{N}(\mu,\sigma^2)}\left[\ell_{exp}(z)\right]$$

Now we can compute

$$\langle\nabla_{w_1}L(w), w_1\rangle = \mathbb{E}_{x_1}\left[g_\sigma'(w_1^\top x_1)x_1^\top w_1\right]$$
$$= \mathbb{E}_{x_1}\left[g_\sigma'(w_1^\top x_1)w_1^\top x_1\right]$$
$$< 0$$

since $g_\sigma'(\mu)$ and $\mu$ always have opposite signs. $\square$

**Lemma A.3.** *In the same setting as in Lemma A.2, we have that for any $w \in S$,*

$$\langle\nabla_{w_2}L(w), w_2\rangle \geq c\|w_2\|_2^2$$

*for some constant $c$ dependent only on $R, \gamma, \tilde{\sigma}_{min}, \tilde{\sigma}_{max}$.*

This lemma relies on the following bound stating that for $|\mu| > r(\sigma)$, $q_\sigma(\mu) > 0$.

**Lemma A.4.** *Define $r(\sigma)$ as in (A.1). Then for $|\mu| \geq r(\sigma)$, $q_\sigma(\mu) \geq \frac{1}{4}\sigma\ell_{exp}(\mu) > 0$.*

We prove this lemma in Section D.1. We also require the following claim that $q_{\tilde{\sigma}}(w_1\gamma)$ is lower bounded by some positive constant for all $w \in S$.

**Claim A.1.** *Define the function $r(\sigma)$ as in Lemma A.4. Suppose $a \leq \mu \leq R\gamma$. Then for all $w \in S$,*

$$\frac{\partial g_{\tilde{\sigma}}(w_1\gamma)}{\partial \tilde{\sigma}} \geq c_1$$

*for some constant $c_1$ dependent only on $R, \gamma, \tilde{\sigma}_{min}, \tilde{\sigma}_{max}$.*

*Proof of Claim A.1.* We note that $r$ and $q$ satisfy the following properties:

1. $r(\sigma)$ is a monotonically increasing increasing function.

2. $q_\sigma(\mu) = \frac{\partial g_{\tilde{\sigma}}(w_1\gamma)}{\partial \tilde{\sigma}} > 0$ for all $\mu \geq r(\sigma)$. (See Lemma A.4 for proof.)

For arbitrary $\tilde{\sigma} \in [R\tilde{\sigma}_{min}, R\tilde{\sigma}_{max}]$, $\mu \geq a = r(R\tilde{\sigma}_{max})$ ensures that $\mu \geq r(\tilde{\sigma})$ by property 1. By property 2, $q(\tilde{\sigma}, \mu) > 0$. Setting $c_1 = \min_{\tilde{\sigma} \in [R\tilde{\sigma}_{min}, R\tilde{\sigma}_{max}], \mu \in [a, R\gamma]} q_\sigma(\mu)$ finishes the proof. $c_1$ is dependent only on $R, \gamma, \tilde{\sigma}_{min}, \tilde{\sigma}_{max}$. $\qquad\square$

*Proof of Lemma A.3.* Using the fact that $x_1$ is a uni-variate mixture of Gaussians, and therefore $w^\top x$ is itself a mixture of two Gaussians with variance $\tilde{\sigma}^2$, we have

$$\begin{aligned}
L(w) &= \mathbb{E}_{y\sim\{\pm 1\}, z\sim\mathcal{N}(0,\tilde{\sigma}^2)}[l_{exp}(yw_1\gamma + z)] \\
&= \frac{1}{2}(g_{\tilde{\sigma}}(w_1\gamma) + g_{\tilde{\sigma}}(-w_1\gamma)) \\
&= g_{\tilde{\sigma}}(w_1\gamma)
\end{aligned}$$

Now we differentiate with respect to $w_2$ to obtain

$$\begin{aligned}
\nabla_{w_2} L(w) &= \frac{\partial L(w)}{\partial \tilde{\sigma}} \cdot \frac{\partial \tilde{\sigma}}{\partial w_2} \\
&= \frac{\partial g_{\tilde{\sigma}}(w_1\gamma)}{\partial \tilde{\sigma}} \cdot 2\Sigma_2 w_2
\end{aligned}$$

We now use the lower bound on $\frac{\partial g_{\tilde{\sigma}}(w_1\gamma)}{\partial \tilde{\sigma}}$ given by Claim A.1. This gives

$$\langle \nabla_{w_2} L(w), w_2 \rangle \geq 2c_1\tilde{\sigma}_{min}||w_2||_2^2.$$

Setting $c = 2c_1\tilde{\sigma}_{min}$ finishes the proof. $\qquad\square$

We can now complete the proof of Theorem 3.2.

*Proof of Theorem 3.2.* Define $\tilde{w}^{t+1} = w^t - \eta\nabla L(w^t)$. By Lemma A.1, $w_1^0\gamma \geq a$. Note that by assumption $a > r(R\tilde{\sigma}_{max})$. By Lemma A.2 and A.3, at iteration $t \geq 0$, taking constant step size $\eta$,

$$\begin{aligned}
|\tilde{w}_1^{t+1}| &> |w_1^t| \\
||\tilde{w}_2^{t+1}||_2^2 &= ||w_2^t - \eta\nabla_{w_2} L(w)|_{w=w^t}||_2^2 \\
&= ||w_2^t||_2^2 + \eta^2||\nabla_{w_2} L(w)|_{w=w^t}||_2^2 \\
&\quad - 2\eta\langle\nabla_{w_2} L(w)|_{w=w^t}, w_2^t\rangle
\end{aligned}$$

By Lemma A.3,

$$\nabla_{w_2} L(w) = q_{\sigma(w)}(\mu(w)) \cdot 2\Sigma_2 w_2$$

for some continuous function $q_\sigma(\mu)$ where $\sigma(w)^2 = w_2^\top\Sigma_2 w_2$, $\mu(w) = w_1\gamma$ over compact set $S$. Therefore $q$ is bounded. Suppose $|q| \leq c_2$ for all $w \in S$, then

$$||\nabla_{w_2} L(w)|_{w=w^t}||_2^2 \leq 4c_2^2\tilde{\sigma}_{max}^2||w_2^t||_2^2.$$

Therefore $||\tilde{w}_2^{t+1}||_2^2 \leq c'||w_2^t||_2^2$ for some constant $c' < 1$ for appropriate choice of $\eta$. As $|\tilde{w}_1^{t+1}| > |w_1^t|$, renormalization results in some constant factor decrease in $||w_2||_2$. Therefore $||w_2^t||_2^2 \leq \epsilon$ when $t \geq K = O(\log(\frac{1}{\epsilon}))$. $\qquad\square$

# B Missing Proofs for Theorem 3.1

Define the following function depending on parameters $\rho, \nu$ defined later:

$$\widetilde{\kappa}(\rho, \nu) \triangleq \min\left\{ \frac{\sqrt{\pi}}{4\sqrt{\rho}}(p^{\star}(\rho,\nu))^{1-\frac{\nu}{4\rho}}\left(\frac{\nu}{2\sqrt{\pi}}\right)^{\frac{\nu}{4\rho}}, \frac{\sqrt{\nu}}{8\sqrt{2\pi}(\sqrt{\rho}+\sqrt{2})}\exp\left(-\left(\frac{\sqrt{\rho}+4}{2\sqrt{\nu}}\right)^2\right)\right\} \tag{B.1}$$

for

$$p^{\star}(\rho,\nu) \triangleq \frac{\sqrt{\nu}}{2\sqrt{\pi}}\min\left\{1, \frac{\sqrt{\nu}}{\sqrt{\rho}}\left(\frac{\sqrt{\pi}}{44\sqrt{2\rho}}\right)^{\frac{8\rho}{\nu}}\right\}$$

Now we choose the constant $\kappa$ in Assumptions 3.1 and 3.2 to be $\kappa(\beta,\alpha) \triangleq \min_{a\in[1,4]}\widetilde{\kappa}(a\beta, a\alpha)$.

Throughout the proof, we will use $p$ to refer to the density of $\mu = w_1^\top x_1$, and define $s(\mu) \triangleq \frac{\partial}{\partial\mu}\log p(\mu) = \frac{p'(\mu)}{p(\mu)}$. Throughout our proofs, we use $\nu, \rho$ to refer to parameters such that $p$ is $\nu$-log-concave and $\rho$-log-smooth. By Assumption 3.1, we have that the density of $\frac{w_1^\top x_1}{\|w_1\|_2}$ is $\alpha$-log-concave and $\beta$-log-smooth, so we can choose $\nu = \alpha/\|w_1\|_2^2$ and $\rho = \beta/\|w_1\|_2^2$ by the linear transformation formula of a probability density. We use $\sigma = w_2^\top \Sigma_2 w_2$ to be the variance of the output of the current classifier restricted to the spurious coordinates.

## B.1 Proof overview

We will argue that if the initial conditions in Assumption 3.2 hold, then they will continue to hold throughout training. Furthermore, under these initial conditions, the loss gradient will force $\|w_2\|_2$ to decrease.

The following three lemmas which analyze a single update of the algorithm will form the main technical core of our proof. They will be used to show that when the loss is sufficiently small, the norm of $w_2$ is always decreasing. The first lemma states that if the loss is small, then $p$ cannot have a large density at 0.

**Lemma B.1.** *In the setting of Theorem 3.1, suppose that $K = 1$. When $L(w) \leq \widetilde{\kappa}(\rho,\nu)$, we must have*

$$L(w) \geq \frac{\sqrt{\pi}}{4\sqrt{\rho}}p(0)^{1-\frac{\nu}{4\rho}}\left(\frac{\sqrt{\nu}}{2\sqrt{\pi}}\right)^{\frac{\nu}{4\rho}} \tag{B.2}$$

*As a consequence, when $L(w) \leq \widetilde{\kappa}(\rho,\nu)$, we have:*

$$p(0) \leq p^{\star}(\rho,\nu)$$

Next, observe that $w_2$ will decrease if $\frac{\partial}{\partial\sigma}L(w) > 0$. The following lemma lower bounds $\frac{\partial}{\partial\sigma}L(w) > 0$ in terms of $p(0)$, showing that if $p(0)$ is small, $\frac{\partial}{\partial\sigma}L(w)$ will be positive.

**Lemma B.2.** *In the setting of Lemma B.1, we have*

$$\frac{\partial}{\partial\sigma}L(w) \geq p(0)\sigma\left(\frac{\sqrt{\pi}}{11\sqrt{\rho}}\left(\frac{\sqrt{\nu}}{2p(0)\sqrt{\pi}}\right)^{\frac{\nu}{4\rho}} - 2\sqrt{2}\max\left\{1, \left(\frac{\sqrt{\rho}}{2p(0)\sqrt{\pi}}\right)^{\frac{\nu}{8\rho}}\right\}\right) \tag{B.3}$$

Finally, this next lemma combines the two lemmas above to show that when the loss is sufficiently small, $w_2$ is always shrinking.

**Lemma B.3.** *In the setting of Lemma B.1, when $L(w) \leq \widetilde{\kappa}(\rho,\nu)$ for $\widetilde{\kappa}(\rho,\nu)$ defined in* (B.1), *we have*

$$\frac{\partial}{\partial\sigma}L(w) \geq \sigma p(0)^{1-\frac{\nu}{4\rho}}\frac{\sqrt{\pi}}{22\sqrt{\rho}}\left(\frac{\sqrt{\nu}}{2\sqrt{\pi}}\right)^{\frac{\nu}{4\rho}} > 0$$

*Proof of Lemma B.3.* Define $a_1 \triangleq \frac{\sqrt{\pi}}{11\sqrt{\rho}} \left( \frac{\sqrt{\nu}}{2p(0)\sqrt{\pi}} \right)^{\frac{\nu}{4\rho}}$, $a_2 \triangleq 2\sqrt{2} \max \left\{ 1, \left( \frac{\sqrt{\rho}}{2p(0)\sqrt{\pi}} \right)^{\frac{\nu}{8\rho}} \right\}$, so that the right hand side of (B.3) becomes $p(0)\sigma(a_1 - a_2)$.

Now we apply Lemma B.1 to conclude that when $L(w) \leq \widetilde{\kappa}(\rho, \nu)$, $p(0) \leq p^\star(\rho, \nu)$. Note that when $p(0) \leq \frac{\sqrt{\nu}}{2\sqrt{\pi}} \min \left\{ \left( \frac{\sqrt{\pi}}{44\sqrt{2\rho}} \right)^{\frac{4\rho}{\nu}}, \frac{\sqrt{\nu}}{\sqrt{\rho}} \left( \frac{\sqrt{\pi}}{44\sqrt{2\rho}} \right)^{\frac{8\rho}{\nu}} \right\}$, we must have $a_1 \geq 2a_2$ by the definitions of $a_1, a_2$. Furthermore, the r.h.s. of this bound is lower-bounded by $p^\star(\rho, \nu)$. As a result, when $p(0) \leq p^\star$, by Lemma B.2, we have $\frac{\partial}{\partial\sigma}L(w) \geq p(0)\sigma\frac{a_1}{2}$. Applying the definition of $a_1$ gives the desired result. $\qquad \square$

## B.2 Proof of Lemmas B.1

The following claim will be useful for proving both Lemma B.1 and Lemma B.2.

**Claim B.1.** *Recall that we defined* $s(\mu) = \frac{\partial}{\partial\mu} \log p(\mu) = \frac{p'(\mu)}{p(\mu)}$. *The following bound holds:*

$$\int_0^\infty \exp\left( \frac{|s(0)|}{2}\delta - \frac{\rho}{2}\delta^2 \right) d\delta \geq \frac{\sqrt{\pi}}{\sqrt{\rho}} \left( \frac{\sqrt{\nu}}{2p(0)\sqrt{\pi}} \right)^{\frac{\nu}{4\rho}}$$

*Proof of Claim B.1.* From Lemma D.3, we start with

$$\int_0^\infty \exp\left( \frac{|s(0)|}{2}\delta - \frac{\rho}{2}\delta^2 \right) d\delta \geq \frac{\sqrt{\pi}\exp\left( \frac{s(0)^2}{4\rho} \right)}{\sqrt{\rho}}$$

Now we apply the lower bound $s(0)^2 \geq \nu \log \left( \frac{\sqrt{\nu}}{2p(0)\sqrt{\pi}} \right)$ and obtain

$$\int_0^\infty \exp\left( \frac{|s(0)|}{2}\delta - \frac{\rho}{2}\delta^2 \right) d\delta \geq \frac{\sqrt{\pi}}{\sqrt{\rho}} \exp\left( \frac{\nu}{4\rho} \log \left( \frac{\sqrt{\nu}}{2p(0)\sqrt{\pi}} \right) \right)$$

$$\geq \frac{\sqrt{\pi}}{\sqrt{\rho}} \left( \frac{\sqrt{\nu}}{2p(0)\sqrt{\pi}} \right)^{\frac{\nu}{4\rho}}$$

$\qquad\qquad \square$

Our starting point is to first lower bound $L(w)$ in terms of $p(0)$ and $s(0)$.

**Claim B.2.** *The following lower bound on the loss $L(w)$ holds:*

$$L(w) \geq 0.25p(0) \int_0^\infty \exp\left( (|s(0)|-1)\delta - \frac{\rho}{2}\delta^2 \right) d\delta$$

*Proof of Claim B.2.* Without loss of generality, assume that $s(0) \geq 0$ (otherwise, by symmetry of $\ell_{exp}$ the same arguments hold). Then we have

$$
\begin{aligned}
L(w) &= \int_{-\infty}^\infty p(\delta)g_\sigma(\delta)d\delta \\
&\geq p(0) \int_{-\infty}^\infty \exp\left( s(0)\delta - \frac{\rho}{2}\delta^2 \right) g_\sigma(\delta) && \text{(by log-smoothness)} \\
&\geq 0.25p(0) \int_{-\infty}^\infty \exp\left( s(0)\delta - \frac{\rho}{2}\delta^2 \right) \ell_{exp}(\delta)d\delta && \text{(by Lemma D.1)} \\
&\geq 0.25p(0) \int_0^\infty \exp\left( (|s(0)|-1)\delta - \frac{\rho}{2}\delta^2 \right) d\delta && \text{(substituting } \ell_{exp}(\delta) = \exp(-\delta) \text{ for } \delta \geq 0)
\end{aligned}
$$

Now the loss is symmetric around 0, so the same argument would also work for $s(0) < 0$. Thus, we obtain the desired result. $\qquad \square$

Next, we argue that if $L(w)$ is bounded above by some threshold, then $s(0)$ will be large in absolute value.

**Claim B.3.** *Suppose that our classifier $w$ satisfies the following loss bound:*

$$L(w) \leq \frac{\sqrt{\nu}}{8\sqrt{2\pi}(\sqrt{\rho} + \sqrt{2})} \exp\left(-\left(\frac{\sqrt{\rho} + 4}{2\sqrt{\nu}}\right)^2\right) \tag{B.4}$$

*Then $|s(0)| \geq \frac{\sqrt{\rho}}{2} + 2$.*

*Proof of Claim B.3.* Assume for the sake of contradiction that $|s(0)| \leq \frac{\sqrt{\rho}}{2} + 2$. First, we consider the case when $s(0) \in [1, \frac{\sqrt{\rho}}{2} + 2]$. In this case, by Lemma D.3, we have

$$p(0) \geq \frac{\sqrt{\nu}}{2\sqrt{\pi}} \exp\left(-\left(\frac{\sqrt{\rho} + 4}{2\sqrt{\nu}}\right)^2\right)$$

Furthermore, in this case we also have $|s(0)| - 1 > 0$, so we can apply (D.7) from Claim D.2. Plugging into Claim B.2, we obtain

$$L(w) \geq 0.25 p(0) \frac{\sqrt{\pi} \exp\left(\frac{(|s(0)| - 1)^2}{\rho}\right)}{\sqrt{\rho}}$$

$$\geq \frac{\sqrt{\nu}}{8\sqrt{\rho}} \exp\left(-\left(\frac{\sqrt{\rho} + 4}{2\sqrt{\nu}}\right)^2\right)$$

In the other case where $0 \leq |s(0)| \leq 1$, by Claim D.2 and Claim B.2, we first have

$$L(w) \geq 0.25 \frac{\sqrt{\pi}}{\sqrt{\rho}} p(0) \exp\left(\frac{(|s(0)| - 1)^2}{\rho}\right) \left(\mathrm{erf}\left(\frac{s(0) - 1}{\sqrt{\rho}}\right) + 1\right)$$

Now applying the lower bound on $p(0)$ from Lemma D.3, we have

$$L(w) \geq \frac{\sqrt{\nu}}{8\sqrt{\rho}} \exp(-s(0)^2/\nu) \exp\left(\frac{(|s(0)| - 1)^2}{\rho}\right) \left(\mathrm{erf}\left(\frac{s(0) - 1}{\sqrt{\rho}}\right) + 1\right)$$

Now by Claim D.3, we have

$$\mathrm{erf}\left(\frac{|s(0)| - 1}{\sqrt{\rho}}\right) + 1 \geq \frac{1}{\sqrt{\pi}} \frac{\exp\left(-\frac{(|s(0)| - 1)^2}{\rho}\right)}{\sqrt{2} + 2(1 - |s(0)|)/\sqrt{\rho}}$$

Thus, we have

$$L(w) \geq \frac{\sqrt{\nu}}{8\sqrt{\pi}(\sqrt{2\rho} + 2(1 - |s(0)|))} \exp\left(-\frac{s(0)^2}{\nu}\right)$$

$$\geq \frac{\sqrt{\nu}}{8(\sqrt{2\pi\rho} + 2\sqrt{\pi})} \exp(-1/\nu)$$

Combining the two cases allows us to conclude that if $|s(0)| < \frac{\sqrt{\rho}}{2} + 2$, the loss must satisfy

$$L(w) \geq \min\left\{\frac{\sqrt{\nu}}{8\sqrt{\rho}} \exp\left(-\left(\frac{\sqrt{\rho} + 4}{2\sqrt{\nu}}\right)^2\right), \frac{\sqrt{\nu}}{8(\sqrt{2\pi\rho} + 2\sqrt{\pi})} \exp(-1/\nu)\right\}$$

Now we note that the r.h.s. of the above equation is lower bounded by the r.h.s of (B.4). Thus, the loss would violate (B.4), a contradiction. □

*Proof of Lemma B.1.* First, by Claim B.3, when $L(w) \leq \widetilde{\kappa}(\rho, \nu)$, we must have $|s(0)| \geq \frac{\sqrt{\rho}}{2} + 2$. Now we lower bound the loss in terms of $p(0)$. Starting from Claim B.2, we have

$$L(w) \geq 0.25 p(0) \int_0^\infty \exp\left((|s(0)| - 1)\delta - \frac{\rho}{2}\delta^2\right) d\delta$$

$$\geq 0.25 p(0) \int_0^\infty \exp\left(\frac{|s(0)|}{2}\delta - \frac{\rho}{2}\delta^2\right) d\delta \qquad \text{(since } |s(0)| \geq 2\text{)}$$

Now applying Claim B.1, we obtain

$$L(w) \geq 0.25 \frac{\sqrt{\pi}}{\sqrt{\rho}} p(0) \left( \frac{\sqrt{\nu}}{2p(0)\sqrt{\pi}} \right)^{\frac{\nu}{4\rho}}$$

$$\geq 0.25 \frac{\sqrt{\pi}}{\sqrt{\rho}} p(0)^{1-\frac{\nu}{4\rho}} \left( \frac{\sqrt{\nu}}{2\sqrt{\pi}} \right)^{\frac{\nu}{4\rho}}$$

This completes the first part of the lemma. For the second part, we note that if $L(w) \leq \widetilde{\kappa}(\rho, \nu)$, then $L(w)$ is bounded above by the r.h.s. of (B.2) with $p^\star(\rho, \nu)$ substituted for $p(0)$ by the definition of $\widetilde{\kappa}(\rho, \nu)$. Combined with the first part of the lemma, this immediately gives $p(0) \leq p^\star(\rho, \nu)$. □

## B.3 Proof of Lemma B.2

We rely on the following lemma which lower bounds $\frac{\partial}{\partial \sigma} L(w)$.

**Lemma B.4.** *Suppose $\sigma \leq \frac{4\sqrt{2}}{\sqrt{\pi}}$ satisfies $\gamma_\sigma^\star \leq \min\left\{ 1, \frac{1}{4\sqrt{\rho}} \right\}$ for $\gamma_\sigma^\star \triangleq \sigma^2 + \sigma \sqrt{2 \log \frac{4\sqrt{2}}{\sqrt{\pi}\sigma}}$. Then the following lower bound on the derivative $\frac{\partial}{\partial \sigma} L(w)$ holds:*

$$\frac{\partial}{\partial \sigma} L(w) \geq$$
$$p(0) \left( \frac{\sigma}{11} \int_0^\infty \exp\left( \left( |s(0)| - \frac{\sqrt{\rho}}{4} - 1 \right) \delta - \frac{\rho}{2} \delta^2 \right) d\delta - 2\sqrt{2} \exp\left( \frac{s(0)^2}{\nu + \sigma^{-2}} \right) \frac{1}{\sqrt{\nu + \sigma^{-2}}} \right) \tag{B.5}$$

*Proof of Lemma B.4.* We compute $\frac{\partial}{\partial \sigma} L(w)$ in two parts:

$$\frac{\partial}{\partial \sigma} L(w) = \int_{|\mu| \leq \gamma_\sigma^\star} p(\mu) \frac{\partial}{\partial \sigma} g_\sigma(\mu) + \int_{|\mu| > \gamma_\sigma^\star} p(\mu) \frac{\partial}{\partial \sigma} g_\sigma(\mu) \tag{B.6}$$

For $|\mu| \leq \gamma_\sigma^\star$, we lower bound the integral using (D.2) in Lemma D.1. For $|\mu| > \gamma_\sigma^\star$, we lower bound the integral using Lemma A.4. By (D.2), we have

$$\int_{|\mu| \leq \gamma_\sigma^\star} p(\mu) \frac{\partial}{\partial \sigma} g_\sigma(\mu) \geq -\sqrt{\frac{2}{\pi}} \int_{|\mu| \leq \gamma_\sigma^\star} p(\mu) \exp\left( -\frac{\mu^2}{2\sigma^2} \right) d\mu$$

$$\geq -\sqrt{\frac{2}{\pi}} \int_{|\delta| \leq \gamma_\sigma^\star} p(0) \exp\left( s(0)\delta - \frac{\nu}{2} \delta^2 \right) \exp\left( -\frac{\delta^2}{2\sigma^2} \right) d\delta$$
$$\text{(by log-strong concavity)}$$

$$\geq -p(0) \sqrt{\frac{2}{\pi}} \int_{-\infty}^\infty \exp\left( s(0)\delta - \left( \frac{\nu}{2} + \frac{1}{2\sigma^2} \right) \delta^2 \right) d\delta$$

$$= -p(0) 2\sqrt{2} \exp\left( \frac{s(0)^2}{\nu + \sigma^{-2}} \right) \frac{1}{\sqrt{\nu + \sigma^{-2}}} \tag{B.7}$$

We obtained the last equation via Claim D.2. Now we lower bound the second integral in (B.6). By Lemma A.4, $\frac{\partial}{\partial \sigma} g_\sigma(\mu) \geq \frac{\sigma}{4} \ell_{exp}(\mu) > 0$ for $|\mu| > \gamma_\sigma^\star$. Assume without loss of generality that $s(0) > 0$ (so we restrict our attention to $\mu > \gamma_\sigma^\star > 0$). By symmetry, our arguments still hold if $s(0) < 0$. Now we have

$$\int_{|\mu| > \gamma_\sigma^\star} p(\mu) \frac{\partial}{\partial \sigma} g_\sigma(\mu) > \int_{\delta > 0} p(\gamma_\sigma^\star + \delta) \frac{\partial}{\partial \sigma} g_\sigma(\gamma_\sigma^\star + \delta)$$

$$\geq \frac{\sigma}{4} \int_{\delta > 0} p(\gamma_\sigma^\star + \delta) \ell_{exp}(\gamma_\sigma^\star + \delta)$$

$$\geq \frac{p(0)\sigma}{4} \int_{\delta > 0} \exp\left( s(0)(\gamma_\sigma^\star + \delta) - \frac{\rho}{2} (\delta + \gamma_\sigma^\star)^2 \right) \exp(-\delta - \gamma_\sigma^\star) d\delta \tag{B.8}$$

Now we note that for $\gamma_\sigma^\star$ satisfying $\sqrt{\rho}\gamma_\sigma^\star \leq \frac{1}{4}$ and $\delta > 0$, we have

$$s(0)(\gamma_\sigma^\star + \delta) - \frac{\rho}{2}(\delta + \gamma_\sigma^\star)^2 > s(0)\delta - \frac{\rho}{2}\delta^2 - \rho\delta\gamma_\sigma^\star - \frac{\rho}{2}\gamma_\sigma^{\star 2} \geq \left(s(0) - \frac{\sqrt{\rho}}{4}\right)\delta - \frac{\rho}{2}\delta^2 - \frac{1}{32}$$

As a result, plugging this back into (B.8) gives

$$\int_{|\mu| > \gamma_\sigma^\star} p(\mu)\frac{\partial}{\partial\sigma}g_\sigma(\mu) > \exp\left(-\gamma_\sigma^\star - \frac{1}{32}\right)\frac{p(0)\sigma}{4}\int_0^\infty \exp\left(\left(s(0) - \frac{\sqrt{\rho}}{4} - 1\right)\delta - \frac{\rho}{2}\delta^2\right)d\delta \tag{B.9}$$

Now we use the fact that $\gamma_\sigma^\star \leq 1$ to lower bound $\exp(-\gamma_\sigma^\star)$. Finally, we obtain (B.5) by combining (B.7) and (B.9). $\qquad\square$

Now we complete the proof of Lemma B.2.

*Proof of Lemma B.2.* Now we proceed to lower bound $\frac{\partial}{\partial\sigma}L(w)$. Our starting point is Lemma B.4. We will lower bound the first integral:

$$\int_0^\infty \exp\left(\left(|s(0)| - \frac{\sqrt{\rho}}{4} - 1\right)\delta - \frac{\rho}{2}\delta^2\right)d\delta \geq \int_0^\infty \exp\left(\left(\frac{|s(0)|}{2}\right)\delta - \frac{\rho}{2}\delta^2\right)d\delta$$

$$\text{(using } |s(0)| \geq \frac{\sqrt{\rho}}{2} + 2\text{)}$$

$$\geq \frac{\sqrt{\pi}}{\sqrt{\rho}}\left(\frac{\sqrt{\nu}}{2p(0)\sqrt{\pi}}\right)^{\frac{\nu}{4\rho}} \qquad \text{(from Claim B.1)}$$

Applying this with equation (B.5) in Lemma B.4, we obtain

$$\frac{\partial}{\partial\sigma}L(w) \geq p(0)\left(\frac{\sigma\sqrt{\pi}}{11\sqrt{\rho}}\left(\frac{\sqrt{\nu}}{2p(0)\sqrt{\pi}}\right)^{\frac{\nu}{4\rho}} - 2\sqrt{2}\exp\left(\frac{s(0)^2}{\nu + \sigma^{-2}}\right)\frac{1}{\sqrt{\nu + \sigma^{-2}}}\right) \tag{B.10}$$

Now we lower bound the second term in (B.10). By applying the upper bound on $s(0)$ in Lemma D.3, we obtain

$$\exp\left(\frac{s(0)^2}{\nu + \sigma^{-2}}\right) \leq \exp\left(\frac{\rho}{\nu + \sigma^{-2}}\log\left(\frac{\sqrt{\rho}}{2p(0)\sqrt{\pi}}\right)\right)$$

$$\leq \left(\frac{\sqrt{\rho}}{2p(0)\sqrt{\pi}}\right)^{\frac{\rho}{\nu + \sigma^{-2}}}$$

Plugging this back into (B.10) and observing that $\frac{1}{\sqrt{\nu + \sigma^{-2}}} \leq \sigma$, we obtain

$$\frac{\partial}{\partial\sigma}L(w) \geq p(0)\sigma\left(\frac{\sqrt{\pi}}{11\sqrt{\rho}}\left(\frac{\sqrt{\nu}}{2p(0)\sqrt{\pi}}\right)^{\frac{\nu}{4\rho}} - 2\sqrt{2}\left(\frac{\sqrt{\rho}}{2p(0)\sqrt{\pi}}\right)^{\frac{\rho}{\nu + \sigma^{-2}}}\right)$$

Now suppose that the condition $\sigma^2\rho^2/\nu \leq 1/8$ holds. Then $\frac{\rho}{\nu + \sigma^{-2}} < \frac{\nu}{8\rho}$, so $\left(\frac{\sqrt{\rho}}{2p(0)\sqrt{\pi}}\right)^{\frac{\rho}{\nu + \sigma^{-2}}} \leq \max\left\{1, \left(\frac{\sqrt{\rho}}{2p(0)\sqrt{\pi}}\right)^{\frac{\nu}{8\rho}}\right\}$. It follows that

$$\frac{\partial}{\partial\sigma}L(w) \geq p(0)\sigma\left(\frac{\sqrt{\pi}}{11\sqrt{\rho}}\left(\frac{\sqrt{\nu}}{2p(0)\sqrt{\pi}}\right)^{\frac{\nu}{4\rho}} - 2\sqrt{2}\max\left\{1, \left(\frac{\sqrt{\rho}}{2p(0)\sqrt{\pi}}\right)^{\frac{\nu}{8\rho}}\right\}\right)$$

$$\square$$

### B.4 Proof of Theorem 3.1

We first show that the loss must be lower-bounded by some constant depending only on the distribution over $x_1$ if $w_1$ is bounded.

**Lemma B.5.** *For any $w$ with $\|w_1\|_2 \leq R$, as long as $\sigma \leq 1$ the following holds:*

$$L(w) \geq 0.25 \exp(-R\mathbb{E}_{x_1}[\|x_1\|_2])$$

*Proof.* We have

$$
\begin{aligned}
L(w) &= \mathbb{E}_{x_1}[g_\sigma(w_1^\top x_1)] \\
&\geq \mathbb{E}_{x_1}[g_\sigma(\|w_1\|_2\|x_1\|_2)] && \text{(B.11)} \\
&\geq 0.25\mathbb{E}_{x_1}[\ell_{exp}(\|w_1\|_2\|x_1\|_2)] && \text{(by Lemma D.1)} \\
&\geq 0.25\exp(-R\mathbb{E}_{x_1}[\|x_1\|_2])
\end{aligned}
$$

The last line followed because $\ell_{exp}(\mu) = \exp(-\mu)$ for positive $\mu$. Since $\exp(-\mu)$ is convex, we applied Jensen's inequality. $\qquad\square$

Next, we argue that $p(0)$ must be lower-bounded by some constant depending only on the distribution over $x_1$ if $w_1$ is bounded.

**Lemma B.6.** *There exists some constant $c_1$ which only depends on $\mathbb{E}_{x_1}[\|x_1\|_2], \alpha, \beta$ such that for all $w$ satisfying $\sigma \leq 1/2$ and $1/2 \leq \|w_1\|_2 \leq 1$, we have*

$$p(0) \geq c_1$$

*Proof.* Fix $\bar{\mu} = \log(4/L(w))$. Then note that we must have

$$\int_{-\bar{\mu}}^{\bar{\mu}} p(\mu)d\mu + \int_{|\mu|>\bar{\mu}} p(\mu)\max_{|\mu|>\bar{\mu}} g_\sigma(\mu) \geq L(w).$$

Now note that since $\sigma \leq 1/2$, Lemma B.6 tells us that $g_\sigma(\mu) \leq 2\exp(-\mu)$. Thus, $\max_{|\mu|>\bar{\mu}} g_\sigma(\mu) \leq L(w)/2$. Thus, we obtain

$$\int_{-\bar{\mu}}^{\bar{\mu}} p(\mu)d\mu + \left(1 - \int_{-\bar{\mu}}^{\bar{\mu}} p(\mu)d\mu\right)\frac{L(w)}{2} \geq L(w)$$

This gives

$$\int_{-\bar{\mu}}^{\bar{\mu}} p(\mu)d\mu \geq \frac{L(w)/2}{1 - L(w)/2}$$

Thus, we can conclude that there exists $\mu' \in [-\bar{\mu}, \bar{\mu}]$ such that $p(\mu') \geq \frac{L(w)}{2\bar{\mu}(2-L(w))}$.

Now we apply Lemma D.3 to obtain

$$|s(\mu')| \leq \sqrt{\rho\log\left(\frac{\sqrt{\rho}\bar{\mu}(2-L(w))}{L(w)\sqrt{\pi}}\right)}$$

Now we apply Claim D.1 to conclude that

$$p(0) \geq p(\mu')\exp\left(-|s(\mu')|\bar{\mu} - \frac{\rho}{2}\bar{\mu}^2\right)$$

Now note that $s(\mu'), p(\mu'), \bar{\mu}$ depend only on $L(w)$ which is upper bounded by 1 and lower bounded by some function of $\mathbb{E}_{x_1}[\|x_1\|_2]$ by Lemma B.5. Furthermore, $\rho \in [\beta, 2\beta]$. Thus, $s(\mu'), p(\mu'), \bar{\mu}$ are all upper and lower bounded by some function of $\mathbb{E}_{x_1}[\|x_1\|_2]$. As a result, the same applies to $p(0)$, giving us the desired statement.

$\qquad\square$

*Proof of Theorem 3.1.* We start with proving the case when $K = 1$. First, we note that $w_1^\top\nabla_{w_1}L(w) < 0$ by using the same argument as Lemma A.2. Furthermore, for all $\|w_1\|_2 \in [1/2, 1]$, the upper bound on $\sigma$ required for Lemmas B.1, B.2, and B.3 are all satisfied by Assumption 3.2. Thus, if $L(w^s) \leq \kappa(\beta, \alpha) = \min_{a \in [1,4]} \tilde{\kappa}(a\beta, a\alpha)$, then initially the loss upper bound for Lemmas B.1, B.2, and B.3 are satisfied. Combining this with Lemma B.6, we get that $\frac{\partial}{\partial\sigma}L(w) \geq c_1\sigma$ for the constant $c_1$ defined in Lemma B.6. Furthermore, the loss $L(w)$ is also always decreasing

for sufficiently small choice of step size. As a result, the following invariants hold throughout the optimization algorithm: $\|w_1\|_2$ is non-decreasing, $\sigma$ is non-increasing, and $L(w) \leq \kappa(\beta, \alpha)$. Thus, the initial conditions Lemmas B.1, B.2, and B.3 will always hold, so we can conclude using the same argument as in Lemma A.3 that $\|w_2\|_2$ is always decreasing with rate $c_2\|w_2\|_2$, where $c_2$ is some value depending on $\alpha, \beta$, and the data distribution. This implies that $w_2$ converges to 0, providing the first statement in Theorem 3.1.

Finally, in the case that $K > 1$, we observe that when Assumption 3.2 is satisfied, we must have $L_i(w) \leq \kappa(\beta, \alpha)$ for all $i$, where $L_i(w)$ is the expectation of the loss conditioned on the $i$-th mixture component. Thus, this immediately reduces to the $K = 1$ case. $\qquad\square$

To prove convergence of noisy gradient descent to an approximate local minimum of the objective

$$\min_{\|w_1\|_2 \leq 1} L((w_1, 0)) \tag{B.12}$$

we also assume that $L((w_1, w_2))$ is twice-differentiable, and furthermore there exists $C$ such that $\nabla_{w_1} L((w_1, w_2)), \nabla^2_{w_1} L((w_1, w_2))$ are Lipschitz in $w_2$ for $\|w_2\|_2 \leq C$, $\|w_1\|_2 \leq 1$.

We will first formally define an $(\epsilon, \gamma)$-approximate local minimum of (B.12). Define $P_{w_1^\perp} \triangleq I - \frac{w_1 w_1^\top}{\|w_1\|_2^2}$ to be the projection onto the space orthogonal to $w_1$. Then an $(\epsilon, \gamma)$-approximate local minimum of (B.12) is a point $w_1 : \|w_1\|_2 \leq 1$ satisfying:

1. $\|w_1\|_2 \geq 1 - \epsilon$.
2. $\|P_{w_1^\perp} \nabla_{w_1} L((w_1, 0))\|_2 \leq \epsilon$.
3. $P_{w_1^\perp} \nabla^2_{w_1} L((w_1, 0)) P_{w_1^\perp} - (w_1^\top \nabla_{w_1} L((w_1, 0))) P_{w_1^\perp} \succeq -\gamma I$.

Note that the first condition simply reflects the fact that all true local minima of (B.12) will satisfy $\|w_1\|_2 = 1$ and therefore lie on the unit sphere $\mathbb{S}^{d_1-1}$, as scaling up the weights only decreases the objective. The second two conditions essentially adapt the classical conditions for approximate local minima (see [25, 1]) to the setting where the domain is a Riemannian manifold (in our case, the unit sphere $\mathbb{S}^{d_1-1}$). In particular, they replace the standard gradient and Hessian with the gradient and Hessian on a Riemannian manifold, in the special case when the manifold is the unit sphere. In other words, they capture the intuition in order for $w_1$ to be a local minimizer of the constrained objective, the only local change to $w_1$ which decreases the loss should be increasing its norm.

To conclude convergence to an approximate local minimum, we note that by the argument of [11], there are sufficiently small step size and additive noise such that for any choice of $\epsilon$, the algorithm converges to an $(\epsilon, \gamma)$-approximate local minimizer of the objective $\min_{\|w\|_2 \leq 1} L(w_1, w_2)$ (defined in the same manner) satisfying $\|w_2\|_2 \leq \epsilon$. By the regularity conditions on $L$, this is also a $(C'\epsilon, C'\gamma)$-approximate local minimizer of the purified objective for some $C'$ depending on the regularity conditions.

# C  Proofs for finite sample setting

Given $n$ samples $X_1, ..., X_n$ define the empirical loss $\widehat{L}$ on *unlabeled* data as:

$$\widehat{L}(w) = \frac{1}{n} \sum_{i=1}^{n} \ell_{exp}(w^T X_i)$$

We analyze self-training on the empirical loss, which begins with a classifier $w^{\mathsf{S}}$, and does projected gradient descent on $\widehat{L}$ with learning rate $\eta$.

$$w^0 = w^{\mathsf{S}}$$

$$w^{t+1} = R \frac{w^t - \eta \nabla \widehat{L}(w^t)}{\|w^t - \eta \nabla \widehat{L}(w^t)\|_2}$$

**Recap of analysis in infinite setting**: In the infinite sample case, gradient descent moves in direction $-\nabla L(w)$, and we show that $\|w_2\|_2 \to 0$ as we self-train. If the loss were convex, we could just analyze the minima and show it had the desired property that $\|w\|_2 = 0$. Standard results in convex analysis would then show convergence. Since the loss is non-convex, the core of the proof is to bound certain directional gradients. In particular, we showed that $\langle \nabla_{w1} L(w), w_1 \rangle < 0$, $\langle \nabla_{w2} L(w), w_2 \rangle \geq c_1^2 \|w_2\|_2^2$, and $\|\nabla L(w)\|_2^2 \leq c_2^2 \|w\|_2^2$. Using this, we analyzed the gradient descent iterates and showed that $\|w_2\|_2$ decreased by a multiplicative factor at each step of self-training.

**Finite sample proof overview**: With finite samples, gradient descent instead moves in direction $-\nabla \widehat{L}(w)$ where $\widehat{L}(w)$ is the empirical loss on $n$ samples. Here $w_2$ won't go to exactly 0, but to a very small value: we will show $\|w_2\|_2 \to \tau$ with high probability if we use $\widetilde{O}(1/\tau^4)$ samples. At a high level, we will show a uniform concentration bound which shows that the empirical gradient $\nabla \widehat{L}(w)$ and population gradient $\nabla L(w)$ are close for all $w$ (Lemma C.1). In Theorem C.1, this lets us show that $\langle \nabla_{w1} \widehat{L}(w), w_1 \rangle$, $\langle \nabla_{w2} \widehat{L}(w), w_2 \rangle$, and $\|\nabla \widehat{L}(w)\|_2^2$ are similar to the population versions above with $L(w)$ instead of $\widehat{L}(w)$. We use this to show that $\|w\|_2$ will keep decreasing until $\|w\|_2 \leq \tau$, and will then stabilize and stay below $\|\tau\|$ forever.

**Notation**: To avoid defining too many constants, we use big-O notation in the following sense that is different from the standard computer science usage but common in learning theory proofs. When we use $O(e)$ in an expression, we mean that expression $e$ can be replaced by $c_e e$ for some universal constant $c_e$ that does not depend on *any* problem parameters (like $\delta, \sigma_{\min}, \gamma$, etc)—it is literally just some number like say $67/32$, but explicitly putting the numbers everywhere makes expressions messy. For sub-Gaussian, sub-exponential, we will use notation and standard results from [40], in particular the *norm* of a sub-Gaussian random variable (Definition 2.5.6), equivalent properties of sub-Gaussian random variables (Proposition 2.5.2), the relationship between sub-Gaussian and sub-exponential random variables (Lemma 2.7.6), and Bernstein's inequality for sub-exponential random variables (Theorem 2.8.1).

We define the empirical expectation $\widehat{E}$ for any function $f$ over the $n$ samples $X_1, \ldots X_n$:

$$\widehat{E}[f(X)] = \frac{1}{n} \sum_{i=1}^{n} f(X_i)$$

## C.1  Results

Our first Lemma shows that the empirical gradients $\nabla \widehat{L}(w)$ and population gradient $\nabla L(w)$ are close for all $w$, if the distribution is sub-Gaussian. We will show later that Gaussian distributions and mixtures of $K$ log-concave distributions are indeed sub-Gaussian. Data that is normalized will also satisfy the sub-Gaussian assumption.

**Lemma C.1.** *Let $\pi : \mathbb{R}^d \to \mathbb{R}^{d'}$ be a projection operator with $d' \leq d$, that is $\pi$ is a $d'$-by-$d$ matrix where each row of $\pi$ is orthnormal, and suppose the distribution $X \sim p(x)$ satisfies that $\|X\|_2$ is sub-Gaussian with norm $B$ (equivalently, variance parameter $B^2$). Suppose we choose:*

$$n = \widetilde{O}\Big(\frac{d}{\epsilon^2} R^2 B^2 \log 1/\delta\Big)$$

*Where we hide terms that are* logarithmic *in* $\frac{1}{\epsilon^2}$, $B$, $R$, *and* $d$ *in the big-O here to highlight the prominent terms, but give the full version in the proof. Then, with probability* $\geq 1 - \delta$, *for all $w$ with* $||w||_2 \leq R$, *we have:*

$$|\widehat{E}[l'(w^\top X)\pi(w)^\top \pi(X)] - E[l'(w^\top X)\pi(w)^\top \pi(X)]| \leq \epsilon \qquad \text{(C.1)}$$

*Proof.* We will use a discretization argument. We first show the concentration in Equation C.1 for fixed $w$ using the fact that the distribution is sub-Gaussian and then applying Hoeffding's inequality. We will then construct an $\epsilon$-cover of the $R$-ball (in $\ell_2$ norm) and use union bound so that the concentration holds for each member of the $\epsilon$-cover. Finally, we will show that the concentration holds for all $w$ with $||w||_2 \leq R$. Let $h(w, X) = l'(w^\top X)\pi(w)^\top \pi(X)$.

**Step 1: Concentration for single $w$:** First, we have:

$$\begin{aligned}
|h(w, X)| = |l'(w^\top X)\pi(w)^\top \pi(X)| \\
\leq |\pi(w)^\top \pi(X)| \\
\leq ||\pi(w)||_2 ||\pi(X)||_2 \\
\leq ||w||_2 ||X||_2 \\
\leq R||X||_2
\end{aligned}$$

We want to bound $\widehat{\mathbb{E}}[h(w, X)] - \mathbb{E}[h(w, X)]$. Since $||X||_2$ is sub-Gaussian with norm $B$, $h(w, X)$ is sub-Gaussian with norm $RB$, it then follows that $h(w, X) - E[h(w, X)]$ is a mean 0 sub-Gaussian random variable with norm $2RB$. So the average, $\widehat{E}[h(w, X)] - E[h(w, X)]$ is mean 0 and sub-Gaussian with norm $2RB/\sqrt{n}$. By the sub-Gaussian tail bound, we then get that with probability at least $1 - \delta/5$:

$$|\widehat{\mathbb{E}}[h(w, X)] - \mathbb{E}[h(w, X)]| \leq O\Big(\frac{1}{\sqrt{n}} RB \sqrt{\log(1/\delta)}\Big)$$

To control the RHS to be less than $\epsilon/3$ with probability at least $1 - \delta/5$, it suffices to choose:

$$n = O\Big(\frac{1}{\epsilon^2} R^2 B^2 \log(1/\delta)\Big)$$

Note that this is only for a single $w$.

**Step 2: $\kappa$-covering:** We will now construct a $\kappa$-covering consisting of $M$ vectors $w$. We will want the above inequality to hold for all $M$ vectors—to do this we will apply union bound. More precisely, a standard covering argument tells us that we can choose $M$ with $\log M \leq d \log(1 + (2R)/\kappa)$ and $M$ vectors $w_1, \ldots, w_M$ s.t. for any $w$ with $||w||_2 \leq R$, there exists $w_i$ with $||w_i||_2 \leq R$ and $||w_i - w||_2 \leq \kappa$. By union bound, we have that if we choose:

$$n = O\Big(\frac{1}{\epsilon^2} R^2 B^2 (\log M + \log(1/\delta))\Big)$$

Then for all $w_i$ the empirical concentration holds, that is with probability at least $1 - \delta/5$, for all $w = w_i$:

$$|\widehat{\mathbb{E}}[h(w, X)] - \mathbb{E}[h(w, X)]| \leq \epsilon/3$$

It now remains to choose $\kappa$ so that we can show this for all $w$ (not just $w = w_i$).

**Step 3: Handling all $w$ by hoosing $\kappa$ small:** To extend the result to all $w$ (not just $w = w_i$), we consider arbitrary $w, w'$ with $||w||_2, ||w'||_2 \leq R$ and $||w - w'||_2 \leq \kappa$. We want to show that the difference in their directional derivatives is not too large. More precisely, we would like to show that with probability $\geq 1 - \delta/5$, for all such $w, w'$:

$$|\mathbb{E}[h(w', X)] - \mathbb{E}[h(w, X)]| \leq \epsilon/3 \qquad \text{(C.2)}$$

And similarly for its empirical counterpart, $\widehat{\mathbb{E}}$. This then proves the main claim, because for any $w$ with $||w||_2 \leq R$, we can choose some $w_i$ in the $\kappa$-cover above. We then have:

$$\begin{aligned}
\widehat{\mathbb{E}}[h(w, X)] - \mathbb{E}[h(w, X)] \leq & |\widehat{\mathbb{E}}[h(w, X)] - \widehat{\mathbb{E}}[h(w_i, X)]| \\
& + |\widehat{\mathbb{E}}[h(w_i, X)] - \mathbb{E}[h(w_i, X)]| \\
& + |\mathbb{E}[h(w_i, X)] - \mathbb{E}[h(w, X)]|
\end{aligned}$$

And each of the terms in the RHS will be bounded above by $\epsilon/3$, so the LHS will be bounded above by $\epsilon$. To show Equation C.2, we first write:

$$h(w', X) - h(w, X) = (l'(w'^\top X)\pi(w')^\top \pi(X) - l'(w^\top X)\pi(w')^\top \pi(X))$$
$$+ (l'(w^\top X)\pi(w')^\top \pi(X) - l'(w^\top X)\pi(w)^\top \pi(X))$$

Using Cauchy-Schwarz, and using the fact that $l'(r) \leq 1$ for all $r$ and that $l'$ is 1-Lipschitz, we can show for the first term in the RHS:

$$|l'(w'^\top X)\pi(w')^\top \pi(X) - l'(w^\top X)\pi(w')^\top \pi(X)| \leq \kappa R||X||_2^2$$

And for the second term in the RHS:

$$|l'(w^\top X)\pi(w')^\top \pi(X) - l'(w^\top X)\pi(w)^\top \pi(X)| \leq \kappa||X||_2$$

Combining the above 2 inequalities:

$$|h(w', X) - h(w, X)| \leq \kappa R||X||_2^2 + \kappa||X||_2$$

We will show below that $\mathbb{E}[||X||_2] = O(B)$, $\widehat{\mathbb{E}}[||X||_2] = O(B)$, $\mathbb{E}[||X||_2^2] = O(B^2)$, $\widehat{\mathbb{E}}[||X||_2^2] = O(B^2)$ (for the empirical expectations, this will hold with probability at least $1 - \delta/5$. Assuming this for now, this gives us that it suffices to choose $\kappa$ such that:

$$\frac{1}{\kappa} \geq \frac{1}{\epsilon}[RB^2 + B]$$

In which case, Equation C.2 and its empirical counterpart hold. In total, this means we require $n$ to be:

$$n = O\left(\frac{1}{\epsilon^2}R^2B^2\left(d\log\left[1 + \frac{R^2B^2 + RB}{\epsilon}\right] + \log(1/\delta)\right)\right)$$

Or omitting log terms except in $1/\delta$ (we keep $1/\delta$ to make the dependence on the probability explicit):

$$n = \widetilde{O}\left(\frac{d}{\epsilon^2}R^2B^2\log(1/\delta)\right)$$

**Bounding the norm and norm-squared**: Finally, we bound the expectations of the norm and norm-squared of $X$, which we used above. By taking integrals, since $X$ is sub-Gaussian with norm $B$, we can show that:

$$\mathbb{E}[||X||_2] \leq O(B)$$

$$\mathbb{E}[||X||_2^2] \leq O(B^2)$$

Next, we will like to bound the empirical means of these quantities. Since $||X||_2$ is sub-Gaussian with norm $B$. This means that $||X||_2 - \mathbb{E}[||X||_2]$ is mean 0 and sub-Gaussian with norm $2B$. So for the average of $n$ iid samples, we have that with probability $\geq 1 - \delta$:

$$\widehat{\mathbb{E}}[||X||_2] \leq \mathbb{E}[||X||_2] + O\left(\frac{B}{\sqrt{n}}\sqrt{\log\frac{1}{\delta}}\right)$$

As long as we choose $n \geq O(\log(1/\delta))$, we have with probability at least $1 - \delta/5$:

$$\widehat{\mathbb{E}}[||X||_2] \leq O(B)$$

Squares of sub-Gaussian random variables are sub-exponential, so $||X||_2^2$ is sub-exponential with sub-exponential norm $O(B^2)$. Then, $||X||_2^2 - \mathbb{E}[||X||_2^2]$ is sub-exponential with sub-exponential norm $O(B^2)$. Then by Bernstein's inequality for sub-exponentials, as long as we choose $n \geq O(\log(1/\delta))$, we have with probability at least $1 - \delta/5$:

$$\widehat{\mathbb{E}}[||X||_2^2] \leq \mathbb{E}[||X||_2^2] + O(B^2) \leq O(B^2)$$

$\square$

The main theorem of this section shows that for a sub-Gaussian distribution, if we have bounds on $\langle \nabla_{w1}L(w), w_1 \rangle$, $\langle \nabla_{w2}L(w), w_2 \rangle$, and $||\nabla L(w)||_2^2$ for the *population*, but do entropy minimization on the *empirical samples*, we will still converge with $||w_2|| \leq \tau$. We will later instantiate these bounds for the Gaussian setting and the more general log-concave setting.

**Theorem C.1.** *Suppose that for all $w$, $\langle \nabla_{w1} L(w), w_1 \rangle < 0$, $\langle \nabla_{w2} L(w), w_2 \rangle \geq c_1 \|w_2\|_2^2$, and $\|\nabla L(w)\|_2^2 \leq c_2^2 \|w\|_2^2$, for some $c_1, c_2 > 0$ where $c_1, c_2$ are not a function of $w$. Let $\tau < 0.5$ be the desired norm for the spurious feature $w_2$, that is, we want $\|w_2\|_2 \leq \tau$ after running self-training. Let $\epsilon = O(c_1 \tau^2)$, and choose $n = \widetilde{O}\left(\frac{1}{\epsilon^2} R^2 B^2 \log(1/\delta)\right)$ according to the Lemma C.1 such that with probability $\geq 1 - \delta$, for all $w$ with $\|w\|_2 \leq \max(R, 1)$ the empirical gradients along both $w_1$ and $w_2$ are near the true gradient:*

$$|\widehat{E}[l'(w^\top X) w_1^\top x_1] - E[l'(w^\top X) w_1^\top x_1]| \leq \epsilon$$

$$|\widehat{E}[l'(w^\top X) w_2^\top x_2] - E[l'(w^\top X) w_2^\top x_2]| \leq \epsilon$$

*Then if initially $\|w_2^0\|_2 \leq 0.5$, self-training with step size $\eta = O(\frac{c_1}{c_2^2})$, will converge to $\|w_2\|_2 \leq \tau$. Specifically, if at step $t$, $\|w_2^t\|_2 \geq \tau/2$, then the norm of $w_2$ shrinks by a multiplicative factor and rapidly reduces to less than $\tau/2$:*

$$\|w_2^{t+1}\|_2^2 < \left(1 - O\left(\frac{c_1}{c_2}\right)^2\right) \|w_2^t\|_2^2$$

*Furthermore, once this has happened, the norm stabilizes: if $\|w_2^t\|_2 < \tau/2$, then $\|w_2^{t+1}\|_2 \leq \tau$.*

*Proof.* We note that $\langle \nabla_{w_2} L(w), w_2 \rangle = \mathbb{E}[l'(w^\top X) w_2^\top x_2]$, $\langle \nabla_{w_2} \widehat{L}(w), w_2 \rangle = \widehat{E}[l'(w^\top X) w_2^\top x_2]$, and similarly for $w_1$. So we have for all $w$ with $\|w\|_2 \leq \max(R, 1)$:

$$|\langle \nabla_{w_1} L(w), w_1 \rangle - \langle \nabla_{w_1} \widehat{L}(w), w_1 \rangle| \leq \epsilon \qquad (C.3)$$

$$|\langle \nabla_{w_2} L(w), w_2 \rangle - \langle \nabla_{w_2} \widehat{L}(w), w_2 \rangle| \leq \epsilon \qquad (C.4)$$

**Step 1: Bounding empirical gradients**: The main optimization analysis requires us to bound 3 quantities: $\langle \nabla_{w_2} \widehat{L}(w), w_2 \rangle$, $\langle \nabla_{w_1} \widehat{L}(w), w_1 \rangle$, and $\|\nabla_{w_2} \widehat{L}(w)\|_2^2$, which we first do.

Equation C.4 gives us a bound on the empirical gradient along $w_2$:

$$\langle \nabla_{w_2} \widehat{L}(w), w_2 \rangle \geq c_1 \|w_2\|_2^2 - \epsilon$$

Equation C.3 this gives us a bound on the empirical gradient along $w_1$:

$$\langle \nabla_{w_1} \widehat{L}(w), w_1 \rangle < \epsilon$$

Finally, we bound $\|\nabla_{w_2} \widehat{L}(w)\|_2^2$:

$$\begin{aligned}
\|\nabla_{w_2} \widehat{L}(w)\|_2 &= (\max_{\|v\|_2 \leq 1} \langle v, \nabla_{w_2} \widehat{L}(w) \rangle)^2 \\
&\leq (\max_{\|v\|_2 \leq 1} \langle v, \nabla_{w_2} L(w) \rangle + \epsilon)^2 \\
&= (\|\nabla_{w_2} L(w)\|_2 + \epsilon)^2 \\
&= 2\|\nabla_{w_2} L(w)\|_2^2 + 2\epsilon^2 \\
&\leq 2c_2^2 \|w_2\|_2^2 + 2\epsilon^2
\end{aligned}$$

Where in the first line we used the variational form of the 2-norm, second line we used Equation C.4, in the third line we used the variational form of the 2-norm again, fourth line we used the fact that $(a + b)^2 \leq 2a^2 + 2b^2$, and in the fifth line we used the bound on $\|\nabla_{w_2} L(w)\|_2^2$ in the asssumption of the theorem.

**Step 2: Show $w_2$ decreases and stabilizes**: Our updates involve taking a gradient descent step, and then projecting back to the sphere, $\|w\|_2 \leq R$. Define $\tilde{w}^{t+1} = w^t - \eta \nabla L(w^t)$ to be the iterate before projecting. Then, we have:

$$\begin{aligned}
\|\tilde{w}_2^{t+1}\|_2^2 &= \|w_2^t - \eta \nabla_{w_2} \widehat{L}(w)|_{w=w^t}\|_2^2 \\
&= \|w_2^t\|_2^2 + \eta^2 \|\nabla_{w_2} \widehat{L}(w)|_{w=w^t}\|_2^2 \\
&\quad - 2\eta \langle \nabla_{w_2} \widehat{L}(w)|_{w=w^t}, w_2^t \rangle \\
&\leq (1 + 2\eta^2 c_2^2 - 2\eta c_1)\|w_2^t\|_2^2 + (2\eta^2 \epsilon^2 + 2\eta\epsilon)
\end{aligned}$$

We choose $\eta$ as:

$$\eta = \frac{c_1}{c_2^2}$$

Which gives us:

$$||\tilde{w}_2^{t+1}||_2^2 \leq \left(1 - \frac{1}{2}\frac{c_1^2}{c_2^2}\right)||w_2^t||_2^2 + (2\eta^2\epsilon^2 + 2\eta\epsilon)$$

Since the norm is always non-negative, we note that $c_1^2/c_2^2 \leq 2$. To control the error terms, we choose $\epsilon$ as:

$$\epsilon = \frac{1}{48}c_1\tau^2$$

Then, we get,

$$2\eta^2\epsilon^2 + 2\eta\epsilon \leq \frac{1}{4}\frac{c_1^2}{c_2^2}(\tau/2)^2$$

In other words, if $||w_2^t||_2 \geq \tau/2$, then the norm decreases:

$$||\tilde{w}_2^{t+1}||_2^2 \leq \left(1 - \frac{1}{4}\frac{c_1^2}{c_2^2}\right)||w_2^t||_2^2$$

And if $||w_2^t||_2 < \tau/2$, then the norm stabilizes:

$$||\tilde{w}_2^{t+1}||_2^2 \leq ||w_2^t||_2^2 + \frac{1}{2}(\tau/2)^2 \leq \frac{3}{2}(\tau/2)^2$$

**Step 3: Show $w_1$ does not decrease much**: We have shown that $\tilde{w}_2^{t+1}$ is smaller than $w^t$. Next, we need to deal with the renormalization step to show that $w_2^{t+1}$ is also smaller than $w_2^t$. We will show that $\tilde{w}_1$ cannot decrease by too much, so that after renormalization, the norm of $w_2$ is still decreasing sufficiently. We have:

$$||\tilde{w}_1^{t+1}||_2^2 = ||w_1^t - \eta\nabla_{w_1}\widehat{L}(w)|_{w=w^t}||_2^2$$
$$\geq ||w_1^t||_2^2 - 2\eta\langle\nabla_{w_1}\widehat{L}(w)|_{w=w^t}, w_1^t\rangle$$
$$\geq ||w_1^t||_2^2 - 2\eta\epsilon$$

From our choise of $\eta$ and $\epsilon$, we can show that $\eta\epsilon$ is actually very small. In particular, with some algebra, we can show that

$$2\eta\epsilon < \frac{1}{24}\frac{c_1^2}{c_2^2} \leq \frac{1}{12}$$

In effect, the decrease in the norm of $w_1$ is at least 10 times smaller than the decrease in the norm of $w_2$. Now we note that since at all times $t$, $||w_2^t|| \leq 0.5$, we have $||w_1^t|| \geq ||w_2^t||$. So $w_1$ is larger and decreases by a much smaller amount, which means that after renormalizing $w_2$ still decreases by around the same amount. Formally, with a bit of algebra, we get that if $||w_2^t||_2 \geq \tau/2$, then *after renormalizing*,

$$||w_2^{t+1}||_2^2 \leq \left(1 - \frac{1}{10}\frac{c_1^2}{c_2^2}\right)||w_2^t||_2^2$$

And on the other hand, if $||w_2^t||_2 < \tau/2$ then *after renormalizing*, $||w_2^{t+1}||_2^2 < \tau^2$. This completes the proof.

$\square$

## C.2 Applying the finite sample results

We now instantiate the above Theorem C.1 for the Guassian setting.

*Proof of Theorem 3.2 finite sample guarantee.* By Lemma A.3, $\langle\nabla_{w2}L(w), w_2\rangle \geq O(\tilde{\sigma}_{\min}||w_2||_2^2)$. By Lemma A.2, $\langle\nabla_{w1}L(w), w_1\rangle < 0$. In the proof of Theorem 3.2 we showed $||\nabla L(w)||_2^2 \leq O(\tilde{\sigma}_{\max}^2||w||_2^2)$. Furthermore, $||X||_2$ is sub-Gaussian with norm $O(\gamma + \tilde{\sigma}_{min}\sqrt{d})$. Applying Theorem C.1, we get that if we choose

$$n = \widetilde{O}\left(\frac{1}{\tau^4}\frac{R^2(\gamma + \tilde{\sigma}_{\max}^2 d)}{\tilde{\sigma}_{\min}^2}\log(1/\delta)\right)$$

then after $t$ iterations, if for all $t \geq T$, where

$$T = O\Big(\frac{\tilde{\sigma}_{\max}^2}{\tilde{\sigma}_{\min}^2} \log \frac{R}{\tau}\Big)$$

We will have $\|w_2^t\| \leq \tau$. Furthermore, since $w_1$ is 1-dimensional in this case and is non-negative initially, from the proof of Theorem C.1, $w_1$ after re-normalizing always increases and stays non-negative. As such, $w_1 \geq \sqrt{R^2 - \tau^2}$. $\qquad\square$

Next, we instantiate the theorem for the mixture of $K$ sliced log-concave setting:

*Proof of Theorem 3.1 finite sample guarantee.* In the population case proof of Theorem 3.1, we showed that $\langle \nabla_{w2} L(w), w_2 \rangle \geq O(c_1 \|w_2\|_2^2)$, $\langle \nabla_{w1} L(w), w_1 \rangle < 0$ , and $\|\nabla L(w)\|_2^2 \leq O(c_2^2 \|w\|_2^2)$. Additionally, we note that a mixture of $K$ sliced log-concave distributions is sub-Gaussian. So we get that there exists some $c, c'$ that depends on the distribution, such that if we choose $n \geq c/\tau^4$ samples then after $t$ iterations, for all $T \geq t$, where $T = O(\log(R/\tau))$, we will have $\|w_2^t\| \leq \tau$. $\qquad\square$

# D  Additional Missing Proofs

## D.1  Bounds on $\ell_{exp}$ and derivatives

*Proof of Lemma A.4.*  We can exactly compute the expression

$$
\begin{aligned}
q_\sigma(\mu) &= \frac{\partial g_\sigma(\mu)}{\partial \sigma} \\
&= \frac{1}{2}\exp\left(\frac{\sigma^2}{2}\right)\sigma\left[\exp(\mu)\operatorname{erfc}\left(\frac{\sigma}{\sqrt{2}}+\frac{\mu}{\sqrt{2}\sigma}\right)+\exp(-\mu)\operatorname{erfc}\left(\frac{\sigma}{\sqrt{2}}-\frac{\mu}{\sqrt{2}\sigma}\right)\right] \\
&\quad - \sqrt{\frac{2}{\pi}}\exp\left(-\frac{\mu^2}{2\sigma^2}\right)
\end{aligned}
\tag{D.1}
$$

As $q_\sigma(\mu)$ and $\ell_{exp}$ are both symmetric around 0, we assume w.l.o.g. that $\mu \geq 0$. We first consider $\sigma \leq \frac{4\sqrt{2}}{\sqrt{\pi}}$. Since $\mu \geq \sigma^2$, $\sigma - \mu/\sigma \leq 0$ so $\operatorname{erfc}((\sigma - \mu/\sigma)/\sqrt{2}) \geq 1$. Thus, we have

$$
q_\sigma(\mu) \geq -\sqrt{\frac{2}{\pi}}\exp\left(-\frac{\mu^2}{2\sigma^2}\right)+\frac{1}{2}\sigma\exp\left(-\mu+\frac{\sigma^2}{2}\right)
$$

Note that for $\mu \geq \gamma_\sigma^\star$, we have

$$
\frac{1}{4}\sigma\exp\left(-\mu+\frac{\sigma^2}{2}\right) \geq \sqrt{\frac{2}{\pi}}\exp\left(-\frac{\mu^2}{2\sigma^2}\right)
$$

in which case we can obtain $q_\sigma(\mu) \geq \frac{1}{4}\sigma\exp\left(-\mu+\frac{\sigma^2}{2}\right)$ by rearranging.

Now we consider $\sigma > \frac{4\sqrt{2}}{\sqrt{\pi}}$. Since $\mu \geq 2\sigma^2$, we have $\operatorname{erfc}((\sigma - \mu/\sigma)/\sqrt{2}) \geq 1$ and $-\sqrt{\frac{2}{\pi}}\exp\left(-\frac{\mu^2}{2\sigma^2}\right) \geq -\sqrt{\frac{2}{\pi}}\ell_{exp}(\mu)$. Therefore

$$
q_\sigma(\mu) \geq -\sqrt{\frac{2}{\pi}}\ell_{exp}(\mu)+\frac{1}{2}\sigma\ell_{exp}(\mu) \geq \frac{1}{4}\sigma\ell_{exp}(\mu)
$$

$\square$

**Lemma D.1.**  *For all $\mu$, the following holds:*

$$
q_\sigma(\mu) = \frac{\partial}{\partial \sigma}g_\sigma(\mu) \geq -\sqrt{\frac{2}{\pi}}\exp\left(-\frac{\mu^2}{2\sigma^2}\right)
\tag{D.2}
$$

*Furthermore, for $\sigma \leq 1$, we also have*

$$
g_\sigma(\mu) \geq 0.25\ell_{exp}(\mu)
\tag{D.3}
$$

*Proof.*  To conclude (D.2), we simply use the fact that erfc is always positive, so only the last term in (D.1) can be negative.

To conclude the second statement, assume without loss of generality that $\mu > 0$. We first note that with probability at least 0.68, $1 \geq Z \geq -1$, and additionally, $\ell_{exp}(\mu + \sigma Z) \geq \exp(-\sigma)\ell_{exp}(\mu)$ for $1 \geq Z \geq -1$. When $\sigma \leq 1$, we thus have $g_\sigma \geq 0.25\ell_{exp}(\mu)$.  $\square$

**Lemma D.2.**  *For all $\mu$ and $\sigma \leq 1/2$, the following holds:*

$$
g_\sigma(\mu) \leq 2\ell_{exp}(\mu)
$$

*Proof.*  Without loss of generality, assume that $\mu > 0$. We note that we can upper bound $\ell_{exp}(\mu)$ by the loss function $\exp(-\mu)$. It follows that

$$
\begin{aligned}
g_\sigma(\mu) &\leq \frac{1}{\sqrt{2\pi}}\int_{-\infty}^{\infty}\exp\left(-\mu-\sigma Z-\frac{Z^2}{2}\right)dZ \\
&= \exp(-\mu)\exp(\sigma^2)\sqrt{2}
\end{aligned}
\tag{D.4}
$$

As $\ell_{exp}$ is symmetric around 0, we could also apply the same argument to $\mu < 0$ using $\exp(\mu)$ as the loss upper bound. This gives the desired result.  $\square$

## D.2 Log-concave and smooth densities

**Claim D.1.** *Let $p : \mathbb{R} \to \mathbb{R}$ be any density such that $\log p$ is differentiable, $\nu$-strongly concave, and $\rho$-smooth. Define $s(\mu) \triangleq \frac{\partial}{\partial \mu} \log p$. Then for all $\mu \in \mathbb{R}$, the following hold:*

$$p(\mu + \delta) \geq p(\mu) \exp \left( s(\mu)\delta - \frac{\rho}{2}\delta^2 \right)$$

$$p(\mu + \delta) \leq p(\mu) \exp \left( s(\mu)\delta - \frac{\nu}{2}\delta^2 \right)$$

*Proof.* By strong concavity and smoothness, we have

$$\log p(\mu + \delta) \geq \log p(\mu) + \frac{\partial}{\partial \mu} \log p(\mu)\delta - \frac{\rho}{2}\delta^2$$

$$\log p(\mu + \delta) \leq \log p(\mu) + \frac{\partial}{\partial \mu} \log p(\mu)\delta - \frac{\nu}{2}\delta^2$$

Exponentiating both sides and using the definition of $s$ gives the desired result. $\qquad\square$

**Lemma D.3.** *In the setting of Claim D.1, we have the following upper and lower bounds for $s$ in terms of $p$:*

$$s^2(\mu) \geq \nu \log \left( \frac{\sqrt{\nu}}{2p(\mu)\sqrt{\pi}} \right)$$

$$s^2(\mu) \leq \rho \log \left( \frac{\sqrt{\rho}}{2p(\mu)\sqrt{\pi}} \right)$$

*In other words, by rearranging,*

$$p(\mu) \geq \frac{\sqrt{\nu}}{2\sqrt{\pi}} \exp(-s(\mu)^2/\nu)$$

$$p(\mu) \leq \frac{\sqrt{\rho}}{2\sqrt{\pi}} \exp(-s(\mu)^2/\rho)$$

*Proof.* First, from Claim D.1, we have

$$1 = \int_{-\infty}^{\infty} p(\mu + \delta)d\delta \leq \int_{-\infty}^{\infty} p(\mu) \exp \left( s(\mu)\delta - \frac{\nu}{2}\delta^2 \right)$$

$$= \frac{2\sqrt{\pi}p(\mu)\exp(s(\mu)^2/\nu)}{\sqrt{\nu}}$$

Solving, we obtain

$$s^2(\mu) \geq \nu \log \left( \frac{\sqrt{\nu}}{2p(\mu)\sqrt{\pi}} \right)$$

Likewise, we can use the same reasoning to obtain

$$s^2(\mu) \leq \rho \log \left( \frac{\sqrt{\rho}}{2p(\mu)\sqrt{\pi}} \right)$$

$\qquad\square$

**Claim D.2.** *For any $a \in \mathbb{R}$, $b > 0$, we have*

$$\int_{-\infty}^{\infty} \exp(ax - bx^2) = \frac{\sqrt{2\pi}\exp(a^2/2b)}{\sqrt{b}} \qquad\qquad (D.5)$$

$$\int_{0}^{\infty} \exp(ax - bx^2) = \frac{\sqrt{\pi}\exp(a^2/2b)\left( \mathrm{erf}\left( \frac{a}{\sqrt{2b}} \right) + 1 \right)}{\sqrt{2b}} \qquad\qquad (D.6)$$

*Furthermore, when $a \geq 0$, we additionally have*

$$\int_{0}^{\infty} \exp(ax - bx^2) \in \left[ \frac{\sqrt{\pi}\exp(a^2/2b)}{\sqrt{2b}}, \frac{\sqrt{2\pi}\exp(a^2/2b)}{\sqrt{b}} \right] \qquad\qquad (D.7)$$

*Proof.* Equations (D.5) and (D.6) follow from direct computation. Equation (D.7) follows because for $a \geq 0$, $1 \geq \mathrm{erf}(a/\sqrt{2b}) \geq 0$. $\square$

**Claim D.3.** *For $a < 0$,*

$$1 + \mathrm{erf}(a) > \frac{2}{\sqrt{\pi}} \frac{\exp(-a^2)}{-a + \sqrt{a^2 + 2}} \geq \frac{1}{\sqrt{\pi}(\sqrt{2} - 2a)} \exp(-a^2)$$

*Proof.* We have $1 + \mathrm{erf}(a) = 1 - \mathrm{erf}(-a) = 1 - (1 - \mathrm{erfc}(-a)) = \mathrm{erfc}(-a)$. Now as $-a > 0$, we can apply the lower bound on $\mathrm{erfc}(-a)$ in [41] to obtain the desired result. $\square$

## D.3 Equivalence between pseudo-labeling variant and entropy minimization

**Proposition D.1.** *The pseudo-labeling algorithm above converges to the same solution as the entropy minimization algorithm in* (2.3).

*Proof of Proposition D.1.* We compute

$$
\begin{aligned}
\nabla_w L_{pseudo}^{t+1}(w)|_{w=w^t} &= \nabla_w \mathop{\mathbb{E}}_{x \sim \mathcal{D}_{\mathrm{tg}}} \ell_{exp}(w^\top x, \mathrm{sgn}\,(w^{t\top} x))|_{w=w^t} \\
&= -\mathop{\mathbb{E}}_{x \sim \mathcal{D}_{\mathrm{tg}}} \exp\left(-w^\top x \cdot \mathrm{sgn}\,(w^{t\top} x)\right) \cdot \mathrm{sgn}\,(w^{t\top} x) x|_{w=w^t} \\
&= -\mathop{\mathbb{E}}_{x \sim \mathcal{D}_{\mathrm{tg}}} \exp\left(-w^\top x \cdot \mathrm{sgn}\,(w^\top x)\right) \cdot \mathrm{sgn}\,(w^\top x) x|_{w=w^t} \\
&= \nabla_w L(w)|_{w=w^t}
\end{aligned}
$$

Therefore for all $t \geq 0$, pseudo-labeling algorithm has the same iterate as entropy minimization (2.3). $\square$

# E  Additional experiments and details

## E.1  Colored MNIST

Among 70K MNIST images, we split the source training / source test / target training / target test into 2:1:3:1. The model architecture is 3-layer feed-forward network with hidden layer sizes 128 and 64. For training on source, we use SGD optimizer with learning rate 0.03, momentum 0.9, weight decay 0.002, and always train until convergence.

**Additional construction and training details for 10-way MNIST.** For each source image, with probability $p$, we assign it a weight $w \overset{\text{unif}}{\sim} [0.1k, 0.1k + 0.1)$ when image is digit $k$; with probability $1 - p$, we assign $w \overset{\text{unif}}{\sim} [0, 1)$. Each target image is assigned $w \overset{\text{unif}}{\sim} [0, 1)$. We create two color channels by scaling the gray-scale image with weights $w$ and $1 - w$.

In entropy minimization phase, we perform full gradient descent on target training set, with learning rate 0.03, momentum 0.9, weight decay 0.002, and train for 300 epochs when $p = 0.95$ and 50 epochs when $p = 0.97$.

**Detailed construction of binary colored MNIST.** In this setup, we assign digits 0-4 label 0 and 5-9 label 1. For each gray-scale image, we first draw a Gaussian random variable $\tilde{w} \sim \mathcal{N}(0, (0.5/3)^2)$. In source domain, with probability $p = 0.8$, example with label $k$ is assigned with $w = 0.5 + (2k-1)|\tilde{w}|$; with probability 0.2, $w = 0.5 + \tilde{w}$. In the target domain, we always have $w = 0.5 + \tilde{w}$. We create two color channels by rescaling the original image with weights $w$ and $1 - w$.

For training, we keep all other hyper-parameters the same as the 10-way setting and only reduce the learning rate to 0.003.

**Distribution of predictions conditioned on gray-scale image.** We examine the effect of entropy minimization on each test example in binary MNIST experiment. For each gray-scale test image $x_1$, we draw 1000 $x_2$, i.e., $\tilde{w} \sim N(0, (0.5/3)^2)$, and plot the distribution of logits $f(x_1, x_2)_y - f(x_1, x_2)_{1-y}$ where $y$ is the true label of $x_1$. According to our theory, if the distribution is concentrated around the positive side, entropy minimization would push the distribution to be more concentrated and positive.

Examples classified wrongly by source classifier [5] (because they were on the negative tail of the distribution) can be corrected due to this effect. Figure 3 is an example image where source classifier was wrong before training on target but corrected due to the explanation we provide. Conversely, examples classified right by source classifier (because they happen to be on the positive tail of the distribution) can turn wrong due to entropy minimization (see Figure 4). The success of entropy minimization relies on more examples concentrated on the positive than the negative side, i.e., source classifier has non-trivial target accuracy.

**Distribution of mean activation.** Figure 5 shows the distribution of $f(x_1, \bar{x}_2)$ before and after self-training for the binary MNIST experiment, where $\bar{x}_2$ indicates neutral color ($w = 0.5$). We see that qualitatively, the empirical distribution of $\mu$ has increasing mass far away from 0 throughout self-training, even for a multi-layer network. This is necessary for our theory, as seen in Figure 2.

**Importance of non-trivial source classifier accuracy.** We provide additional details on our study of 10-way colored MNIST when the spurious correlation probability is $p = 0.97$. In this setting, the source classifier has 98% test accuracy on source but only 72% on target. Entropy minimization initialized at $\tilde{f}$ causes target accuracy to *drop* to 67% (see Figure 6 right).

## E.2  CelebA dataset

We partition the celebA dataset [21] so that the source domain has a perfect correlation between gender and hair color: 1250 blond males, 1749 non-blond females. The target domain has 57K unlabeled examples with the same correlation between gender and hair color as in the original dataset (Figure 7).

Figure 3: Distribution of $f(x_1, x_2)_y - f(x_1, x_2)_{1-y}$ before (**left**) and after (**right**) self-training for a test image whose prediction turned from wrong to correct. Green line shows $f(x_1, \widehat{x_2})$ turning positive for the particular $\widehat{x_2}$ in test set.

Figure 4: Distribution of $f(x_1, x_2)_y - f(x_1, x_2)_{1-y}$ before (**left**) and after (**right**) self-training for a test image whose prediction turned from right to wrong. Green line shows $f(x_1, \widehat{x_2})$ turning negative for the particular $\widehat{x_2}$ in test set.

Figure 5: Distribution of $f(x_1, \bar{x_2})$ before (**left**) and after (**right**) self-training across all test images $x_1$ for neutral color $\bar{x_2}$. Qualitatively the empirical distribution of $\mu$ has more mass far away from 0 after self-training (which is the desired case for our theory, as seen in Figure 2).

Figure 6: In the 10-way MNIST experiment, entropy minimization raises target test accuracy by 9% when we initialize with a good source classifier (**left**) and decreases target accuracy when we initialize with a bad source classifier (**right**). **Left**: Spurious correlation in source is $p = 0.95$ so source classifier obtains high target accuracy; **Right**: Spurious correlation in source is $p = 0.97$ so source classifier does not learn the right features.

(a) Synthetic source data: blondness perfectly correlates with the male gender.

(b) Synthetic target data: each gender has a variety of hair colors.

(c) Predictions corrected by self-training were mostly mistaken due to the spurious correlation.

Figure 7: In the synthetic CelebA experiment, the source has perfect correlation between hair color and gender, and the target does not. A classifier trained only on the source domain uses the spurious correlation. However, continuing to self-train on the unlabeled target domain reduces reliance on the spurious feature.

We use entropy minimization on this dataset with the Conv-Small model in [24]. The source classifier has 94% accuracy on source data and 81% on target. After training on the sum of the source labeled loss and target entropy loss, the target accuracy increases to 88%.

### E.3 Connection between entropy minimization and stochastic pseudo-labeling

In Equation **??** we point out that entropy minimization is equivalent to a stochastic version of pseudo-labeling where we update the pseudo-labels after every SGD step. In practice, pseudo-labels are often updated for only a few rounds, and the student model is usually trained to convergence between rounds [42]. In the 10-way MNIST experiment, we perform 3, 6, 30 rounds of pseudo-labeling with 100, 50, and 10 epochs of training per round, interpolating between more common versions of pseudo-labeling and entropy minimization. Figure 8 shows that entropy minimization converges to better target accuracy within the same clock-time, suggesting that practitioners may benefit from pseudo-labeling with more rounds and fewer epochs per round.

### E.4 Toy Gaussian mixture setting

**Generating data.** We generate source examples in the following fashion: For each example $(x_1, x_2) \in \mathbb{R}^4$, we first sample $y$ uniformly from $\{-1, 1\}$, and then $x_1 \in \mathbb{R}^2 \sim \mathcal{N}(\gamma y, I)$, where $\gamma$ is a random 2-dimensional vector. For the source examples, we then sample $\tilde{x}_2 \sim N(\vec{0}, I)$. For each coordinate $i$ of $x_2$ ($i \in \{1, 2\}$), with probability 0.8, we set $(x_2)_i = y|(\tilde{x}_2)_i|$ (correlated); with probability 0.2, we set $(x_2)_i = (\tilde{x}_2)_i$ (uncorrelated). For target examples, we sample $x_2 \sim N(\vec{0}, I)$.

The source training dataset, source test set, and target test set all have 10K examples.

**Algorithms.** We use entropy minimization as well as the following version of pseudo-labeling: starting with the source classifier, we perform 200 rounds of pseudo-labeling with 50 epochs of training in each round. We also set up a threshold $\tau = 0.1$ where we throw out least-confident target

Figure 8: In the 10-way MNIST experiment, 3 rounds of pseudo-labeling with 100 epochs per round (**left**), 6 rounds of pseudo-labeling with 50 epochs per round (**middle**), and 30 rounds of pseudo-labeling with 10 epochs per round (**right**) increase in target accuracy.

example $x$ with $|w^\top x| < \tau$ in each round to mimic most popular pseudo-labeling algorithms used in practice [42]. We experiment on this version of pseudo-labeling algorithm because the version in equation **??** is equivalent to entropy minimization.

For entropy minimization, we use a new batch of 10K target training examples in each epoch. We use SGD optimizer with learning rate 1e-3 and normalize the linear model after each gradient step.

For pseudo-labeling, we use a new batch of 10K target examples in each round. Optimizer choices are the same as entropy minimization.

Figure 9: Entropy minimization (**left**) and pseudo-labeling (**right**) increase target test accuracy from 95.9% to 97.5%, and reduce coefficients on two spurious coordinates from 0.33 to 0 in the Gaussian mixture experiment.

**Improvement of target test accuracy and de-emphasis of spurious features.** In the Gaussian mixture experiment, the source classifier gets an accuracy of 95.9% on the target domain. Both entropy minimization and pseudo-labeling algorithms raise the target accuracy to Bayes-optimal while driving the coefficients $w_2$ on spurious features $x_2$ to 0 (Figure 9). Notably, even though we use confidence thresholding and train for 50 epochs in each round, the model behavior still closely tracks that of entropy minimization, as predicted by our theory.

### E.5 Justification of approximation $l(t) = \exp(-|t|)$

Self-training on target using $\ell(t) = \exp(-|t|)$ as an approximation for $\ell_{ent}(t)$ produces the same effect for binary MNIST (see figure 11). We plot the training loss $l(t) = \exp(-|t|)$ and $l_{ent}(t)$ (Figure 12) to show that they track each other really well.

Figure 10: Plot of $exp(-|t|)$ and entropy loss. The losses are within a constant factor of each other and exhibit the same tail behavior.

Figure 11: Target accuracy using $l(t) = \exp(-|t|)$ on binary MNIST dataset.

Figure 12: **Left**: Training loss using $l(t) = \exp(-|t|)$; **Right**: Entropy loss when training using $l(t) = \exp(-|t|)$.