[Reviews · NeurIPS 2020]

Review 1

Summary and Contributions: This paper concentrates its theoretical study on the utility of self-training (in the form of min-entropy minimization/pseudo-labeling algorithms which do not have access to labels y) in a simple generative model with a structured domain shift. The key idea in the set-up assumes one is given an arbitrary source/initialization classifier $w_S$, and then presented with labels from a target domain in which there are some “useful” features x_1 correlated with labels y, and then spurious features x_2 which are just noise unrelated to the labels y. Under Gaussian assumptions on the spurious features, a suitable mixture assumption on the useful features, and good source initialization guarantees are provided to show the aforementioned algorithms return vectors with small norm on the spurious support. Several theoretical counter-examples/thought experiments and simulated real-data experiments are provided to validate the theory.

Strengths: This paper targets a timely and important problem. With regards to domain shift, I also like that this paper posits/analyzes (albeit a simple) structured model of domain shift to study self-training. Much prior work in distributional robustness assumes a worst-case model for distribution shift which seems too pessimistic to capture behavior that is salient to real applications. I also enjoyed reading the intuitions in the proof sketch and examples/toy experiments used to support the validity of the assumptions; indeed I believe one of the key contributions of the paper is not the proof techniques (which seem to involve some detailed computations to bounds densities of MoG in various regimes for example in some cases) but the rather the relevance of the assumptions and set-up. I think Theorems 3.1 and Theorem 3.2 form a nice contribution showing how some forms of unsupervised self-training can provide benefits in a simple setting.

Weaknesses: One important caveat to the above is that an important missing component of Theorem 3.1 is a bound on the target accuracy of the final classifier (in analogy to Theorem 3.2 which studies a much simpler setting). The algorithmic set-up of pseudo-labeling is nice, but I feel the statistical implications for generalization is an important point in a paper studying self-training -- perhaps more important than the algorithmic connections to min-entropy minimization. Identifying assumptions and conditions on when self-training can aid learning--using any procedure--would seems to be the most interesting contribution. Clarification on why this is not provided or difficult to provide in the context of Theorem 3.1 would be useful. The setting studied is also simple but I do not view this as a significant downside.

Correctness: The claims appear correct.

Clarity: The paper is well-written. One small point is that in the pre-amble to Theorem 3.2 the parameter $\gamma$ is not explained (and is not referenced in Theorem 3.2); it appears related to the SNR for the problem but am not sure. One other point I may have misunderstood is that in the second paragraph of Section 3 it is stated that “... motivates our assumptions of separation … and that the spurious $x_2$ is a mixture of sliced log-concave distributions”. My understanding was that $x_2$ is assumed to be Gaussian (based on Equation 2.1) and $x_1$ is a mixture.

Relation to Prior Work: Seems to be cited appropriately.

Reproducibility: Yes

Additional Feedback: I have read the reviews/feedback and the appreciate the comment on relationship to max-margin classification. I will maintain my score.


Review 2

Summary and Contributions: The paper studies why self-training can avoid spurious features based convergence analysis, which tries to explain the recent empirical success using self-training on large scale data. Under a linear model with binary label and mild assumptions on the distribution of input features, the paper shows that if the initialization is good enough, entropy-based update can penalize the parameter of those spurious features. The theory is validated with synthetic experiments on variants of CelebA and MNIST dataset.

Strengths: It studies the power of self-training from a theoretical perspective and proposes a toy but new model setting to give convergence analysis. Self-training for large-scale data is a hot topic, and this paper tries to provide some explanation.

Weaknesses: While trying to explain the success of self-training, it is unclear how the framework or the theory can be generalized / provide guidance to empirical discoveries. From the theoretical side, the `surprising’ part mentioned in the paper is not that surprising because the loss function would favor good features due to their correlation with the model. === Post rebuttal == Thanks for writing the rebuttal. I have carefully read it and understand the arguments. Based on all the feedback and my own evaluation, I would keep my score as it is.

Correctness: I did not check the proof details but the claims seem to be correct.

Clarity: In general yes -- there are some typos and formatting issue which can be fixed easily.

Relation to Prior Work: I think so.

Reproducibility: Yes

Additional Feedback: I did not check the proof details but I am interested in the following: It seems that Theorem 3.2 holds for d_2 > 1 and the only condition that depends on Sigma_2 is the minimum eigenvalue -- so is the analysis here equivalent to analyzing d_2=1 ?


Review 3

Summary and Contributions: For linear classifiers with certain data distribution assumptions, the authors prove that when the initial classifier is sufficiently accurate, self-training, via either pseudo-labels or entropy minimization, will improve robustness of the classifier against domain shift caused by spurious features. In particular, such improvement occurs due to a feature selection effect. This work provides a novel theoretical explanation on effectiveness of self-training on unsupervised domain adaptation problems.

Strengths: The work makes great efforts in carefully crafting assumptions to lay a workable foundation for the theoretical work. The theoretical results are insightful and the experimental results support their theoretical implications. Its feature selection mechanism can be further examined by practitioner.

Weaknesses: 1. There are many technical assumptions and it is hard to identify which play the essential role in the result and which are just for technical issues, and thus possible to relax in the future. 2. The loss functions for the pseudo-label self-training and entropy minimization are not standard. I suggest analyzing log-loss for the pseudo-label self-training and making the equivalence between l_exp and l_ent formal. 3. The authors need to explain why the additional source loss on labeled data in self-training is required in the experiment on CelebA. === Post rebuttal === The author's response answers my questions. My rate remains unchanged.

Correctness: I think the claims and methods are correct, though the proof is not carefully checked.

Clarity: The paper overall is well written. But several places can be further improved. 1. l_exp is used for two different purpose on Page 3. 2. Instead of showing many but far from sufficient details of the proof in Section 4, providing more intuition and interpretation about the relation between assumptions and conclusions may be more helpful. 3. The proof of Prop 2.1 is in Appendix D not E.

Relation to Prior Work: The relation between this work and previous contributions is clearly discussed.

Reproducibility: Yes

Additional Feedback:


Review 4

Summary and Contributions: The authors propose an analysis fin the setting of unsupervised domain adaptation for avoiding spurious features which correlate with the source labels but have no correlation with the target labels. The authors claim and prove that entropy minimization on unlabelled target data will avoid using the spurious feature if initialized with a decently accurate source classifier, even though the objective is non-convex and contains multiple bad local minima using the spurious features.

Strengths: The paper is well written and all the theoretical claims are well established. Spurious or useless features which do not have any correlation with the target domain are a source of inherent bias for many unsupervised domain adaptation tasks. This paper proves that these inherent bias is automatically avoided given a good initial classifier for the source domain. The authors have also carried out experimental studies on two datasets – a semi synthetic coloured MNIST and CelebA. In both the datasets, the results validate the claims made by showing that self-training with a decent initial classifier indeed improves the performance in the target domain.

Weaknesses: Even though it is understood that the paper is more theoretical in nature, the experimental section feels ad-hoc. The authors assume that the spurious features are Gaussian and the non-spurious, a mixture of log-concave distributions. However, the experiments do not show that these assumptions are hold. Is it possible to show that the weights for the spurious features are indeed minimized to zero while doing the self-supervised pre-training? Also, what happens to the model accuracy if the classifier gets stuck in some sub-optimal point? Will self-training reject low confidence samples? It will be better if there is some study of any effect of regularization on the self-training approach to identify and avoid spurious features. Post-Rebuttal: Concerns addressed in rebuttal. Ratings unchanged, inspite of some minor weaknesses and promises to add some details. Will not object is majority of reviewers feel otherwise for ratings.

Correctness: Appears to be correct

Clarity: Quite clear

Relation to Prior Work: Discussions on prior work done

Reproducibility: Yes

Additional Feedback:

[Author Response · NeurIPS 2020]



(a) Synthetic source data: blondness perfectly correlates with the male gender.

(b) Synthetic target data: each gender has a variety of hair colors.

(c) Samples corrected by self-training were mostly mistakenly predicted because of the spurious correlation.

We thank the reviewers for the detailed and insightful feedback. The reviewers noted that the paper "target[s] a timely
and important problem" and "posits/analyzes a structured model... to study self-training" [R1], "the theoretical claims
are well established" [R4], and "the experimental results support the theoretical implications" [R3]. We will address the
major points and incorporate others in the next revision.

**[R1]:** "missing ... a bound on the target accuracy of the final classifier (in analogy to Theorem 3.2 which studies a
simpler setting). Clarification on why this is not provided or difficult to provide ... would be useful."
• The strongest result obtained, as noted in Line 211-212, is that the final classifier is a min-entropy solution, or max
unsupervised margin solution, **without** using spurious features. Proving this classifier also obtains good accuracy
requires analyzing unsupervised margin maximization, which needs stronger data assumptions[1] and is beyond the scope
of the paper. Unsupervised margin maximization was proposed 25 years ago (as transductive SVM [5]), but has little
theoretical analysis (except [1]). Instead, we focus on removing the spurious features.
**[R1]:** "$\gamma$ is not explained", "$x_2$ is assumed to be Gaussian... and $x_1$ is a mixture"
• $\gamma/\sigma$ is indeed the SNR. The condition that $w_S$ has $1 - \rho$ accuracy is equivalent to requiring that $w_1\gamma$ is large (Lemma
A.1 in Appendix A). $x_2$ should be $x_1$ (a typo) in line 141.
**[R2]:** "[theory not surprising because loss] would favor good features due to their correlation with the model", "unclear
how... the theory can... provide guidance to empirical discoveries"
• We respectfully and strongly disagree. The spurious features are indeed independent with ground-truth target labels,
but **no** target label is provided in self-training. The classifier is entirely self-trained on pseudolabels, which can in fact
be correlated with the spurious features due to biases from the source-trained initial classifier.
• Our theory suggests that self-training on diverse unlabeled datasets can improve model robustness, which guides
practitioners to collect larger and more diverse datasets even if labeling is impractical. Recent empirical results [6]
are consistent with our theory but do not explain why self-training works. We also show entropy minimization can
converge faster than pseudo-labeling (Appendix E.3) which may inform practitioners.
**[R3]:** "the [exponential] loss functions [is] not standard. I suggest analyzing log-loss"
• We would like to respectfully argue that the distinction between the two losses is unimportant in the context of our
paper. In theoretical works, these losses have been regarded as equivalent in various contexts [3, 4], and analyzing
exponential loss is convenient for the proof. In Figure 10, the two losses achieve equivalent empirical performance.
**[R3]:** "hard to identify which [assumptions] play the essential role in the result and which are just for technical issues",
"why the additional source loss on labeled data in self-training is required in the experiment on CelebA"
• In the setting of Theorem 3.1, we make two main assumptions: 1. the signal $x_1$ is structured and 2. the source
classifier is decent. Both assumptions are essential because they rule out failure cases of self-training in Section 3.1.
• The source loss constrains self-training to stay close to the source classifier, ensuring that the classifier correctly uses
signal features while entropy minimization removes the spurious ones. Previous empirical works have found the need
for explicit constraints to stay close to source classifier [2].
**[R4]:** "show that the weights for the spurious features are indeed minimized to zero", "what happens to ... accuracy if
the classifier gets stuck ... Will self-training reject low confidence samples?", "any effect of regularization"
• Figure 8 shows spurious coordinates go to 0 in the Gaussian mixture experiment. For colored MNIST, Figures 3
and 4 show reduced reliance on $x_2$. For our CelebA experiment, visualization of a random sample of examples whose
predictions are corrected by self-training shows that self-training removes the spurious feature (see attached figure).
• The local minima of the min-entropy objective that self-training converges to can have sub-optimal accuracy. We do
not reject any examples in our experiments as our theory does not analyze rejection.
• Regularization is necessary; without it, scaling up the source predictions can make the loss arbitrarily small. Our
theory analyzes l2 regularization by constraining the classifier norm to 1. We will clarify this point in the next revision.
**References:** [1] Derberko et al., Error bounds for transductive learning via compression and clustering, 2004. [2]
Shu et al., A dirt-t approach to unsupervised domain adaptation, 2018. [3] Soudry et al., The implicit bias of gradient
descent on separable data, 2018. [4] Telgarsky, Margins, shrinkage, and boosting, 2013. [5] Vapnik, The nature of
statistical learning theory, 1995. [6] Xie et al., Self-training with noisy student improves imagenet classification, 2020.

## Footnotes

[1]The authors suspect that a difficult situation is when all the data reside on the hypercube $\{-1, +1\}^d$, where a large unsupervised margin classifier may exist coincidentally due to the discrete nature. We need to make careful assumptions to rule out such cases.


[Meta-Review · NeurIPS 2020]

The authors prove that when the initial classifier is sufficiently accurate, self-training, via either pseudo-labels or entropy minimization, will improve robustness of the classifier against domain shift caused by spurious features. It is a piece of solid theory paper. Referees agree that this work is above the threshold and do not have further concerns after reading the authors' rebuttal.